# *Cryptococcus neoformans* adapts to the host environment through TOR-mediated remodeling of phospholipid asymmetry

Laura C. Ristow [1], Andrew J. Jezewski [1], Benjamin J. Chadwick [2], Mark A. Stamnes[3], Xiaorong Lin [2,4] & Damian J. Krysan[1,3,5] ✉

*Cryptococcus* spp. are environmental fungi that first must adapt to the host environment before they can cause life-threatening meningitis in immuno-compromised patients. Host $CO_2$ concentrations are 100-fold higher than the external environment and strains unable to grow at host $CO_2$ concentrations are not pathogenic. Using a genetic screening and transcriptional profiling approach, we report that the TOR pathway is critical for *C. neoformans* adaptation to host $CO_2$ partly through Ypk1-dependent remodeling of phosphatidylserine asymmetry at the plasma membrane. We also describe a *C. neoformans* ABC/PDR transporter (*PDR9*) that is highly expressed in $CO_2$-sensitive environmental strains, suppresses $CO_2$-induced phosphatidylserine/phospholipid remodeling, and increases susceptibility to host concentrations of $CO_2$. Interestingly, regulation of plasma membrane lipid asymmetry by the TOR-Ypk1 axis is distinct in *C. neoformans* compared to *S. cerevisiae*. Finally, host $CO_2$ concentrations suppress the *C. neoformans* pathways that respond to host temperature (Mpk1) and pH (Rim101), indicating that host adaptation requires a stringent balance among distinct stress responses.

*Cryptococcus* spp. cause over one hundred thousand deaths per year across the globe with the majority occurring in those living with HIV[1]. *Cryptococcus* spp. are environmental basidiomycetes that occupy a variety of niches and have broad genetic diversity[2]. However, only a minority of these species cause human disease[3]. Accordingly, the mechanisms and traits that distinguish pathogenic and non-pathogenic *Cryptococcus* spp. have been of intense interest[4]. Three key virulence traits have emerged from these studies: (1) the ability to grow at host body temperature (37 °C); (2) the production of a poly-saccharide capsule; and (3) the generation of melanin pigment[1]. Strains or species that lack these traits are hypo-virulent and, conversely, expression of the three traits correlates with patient outcome among strains that infect humans[5,6].

The big three virulence traits do not, however, completely explain the variations in virulence among *Cryptococcus* spp. and strains[7]. For example, Litvintseva and Mitchell showed that the virulence of *C. neoformans* strains isolated from the environment was greatly reduced relative to clinical strains despite no difference in the expression of the big three traits[3]. In addition, a systematic comparison of virulence traits and outcomes of genetically similar *C. neoformans* strains iso-lated from patients demonstrated wide variation in patient outcomes that could not be explained by differences in high temperature growth, capsule formation, or melaninization[8,9]. These observations clearly indicate that additional virulence traits must distinguish *Cryptococcus* strains that are successful mammalian pathogens from those that are not.

[1]Department of Pediatrics, Carver College of Medicine, University of Iowa, Iowa City, IA 52242, USA. [2]Department of Plant Biology, University of Georgia, Athens, GA 30602, USA. [3]Department of Molecular Physiology and Biophysics, Caver College of Medicine, University of Iowa, Iowa City, IA 52242, USA. [4]Department of Microbiology, University of Georgia, Athens, GA 30602, USA. [5]Microbiology and Immunology, Carver College of Medicine, University of Iowa, Iowa City, IA 52242, USA. ✉e-mail: damian-krysan@uiowa.edu

Our group recently found that *C. neoformans* strains isolated from the environment and unable to infect humans or mammals grow poorly in an atmosphere that contained carbon dioxide ($CO_2$) concentrations (5%) corresponding to the host environment[10]. In contrast, strains isolated from patients were much more tolerant to an atmosphere of 5% $CO_2$. Indeed, the few environmental isolates that grew well in 5% $CO_2$ were also virulent in a mouse model of cryptococcosis. Importantly, the effects of $CO_2$ on the growth of environmental and clinical *C. neoformans* strains were independent of medium pH. The environmental niche of *C. neoformans* contains approximately 0.04% $CO_2$ while most tissues in the human host have $CO_2$ concentrations that correspond to 5%. Therefore, the transition from an environmental niche to the mammalian host requires the fungus to tolerate ~100-fold increase in $CO_2$ concentrations and our previous work indicates that this is a significant, independent host-related stress that the fungus must overcome[10].

To identify the mechanisms that underlie $CO_2$ tolerance in *C. neoformans*, we screened systematic libraries of deletion mutants covering protein kinases and transcription factors in the $CO_2$-tolerant reference strain H99[11,12]. Coupling these screens with transcriptional profiling revealed that $CO_2$ tolerance is dependent upon the target of rapamycin (TOR) pathway and membrane phospholipid asymmetry. Importantly, we have also found evidence that the TOR pathway's role in phospholipid asymmetry has been significantly rewired relative to the model yeast, *Saccharomyces cerevisiae*, and we identified a putative ABC/PDR family transporter, *PDR9*, that plays a key role *C. neoformans* $CO_2$ tolerance. Finally, the TOR-mediated response to host $CO_2$ suppresses the activation of cell pathways that respond to host temperature and pH. As a result, *C. neoformans* cells must strike an exquisite balance between potentially contradicting stress responses to successfully transition from the environment to the host.

## Results

### The TOR and cell wall integrity MAPK pathways play opposite roles during $CO_2$ stress

Protein kinases (PK) and transcription factors (TF) are key regulators of the cellular response to extracellular stresses. To identify PKs and TFs that affect $CO_2$ tolerance, we screened systematic deletion mutant libraries covering the majority of non-essential PKs and TFs in the $CO_2$ tolerant *C. neoformans* strain background H99[11,12]. Host levels of $CO_2$ are fungistatic to intolerant strains and mutants rather than fungicidal[10]. To generate quantitative data on the relative fitness of these mutants, we developed a competitive fitness assay (Fig. 1a). H99 expressing mNeonGreen was used as a reference strain and inoculated into a 96-well microtiter plate with an unlabeled PK or TF mutant at a 1:1 ratio in host-relevant RPMI 1640 medium buffered to pH 7 at 30 °C under either ambient or 5% $CO_2$ for 24 h. The screen was performed at 30 °C rather than at the host temperature of 37 °C so that the $CO_2$ phenotype of temperature sensitive mutants could be assessed. The relative ratio of mutant to reference was determined by flow cytometry and the ratio at 5% $CO_2$ was normalized to growth under ambient air to generate a $CO_2$ fitness score. Control experiments comparing H99 to $CO_2$-intolerant strains demonstrated that the assay identified strains with fitness defects at 5% $CO_2$ (Supplementary Fig. 1a).

We screened a total of 129 PK and 155 TF mutants in duplicate[11,12]. Scatter plots summarizing the screens are shown in Fig. 1b & c. A total of 21 PK mutants showed statistically significant alterations in competitive fitness relative to H99 with 14 mutants hyper-susceptible to $CO_2$ while 7 mutants displayed increased fitness. Importantly, the *cbk1Δ* mutant showed reduced fitness which corroborates Chadwick et al. who showed that the RAM pathway is important for $CO_2$ tolerance[13]. In contrast, all but one of the TF mutants with altered $CO_2$ fitness are more resistant than H99. The full results of the screen are provided in Supplementary Data 1 and 2. We re-constructed 7 PK and 4 TF deletion mutants with altered $CO_2$ fitness and determined their

competitive fitness in independent experiments (Fig. 1d, e). The majority of mutants tested on agar plates showed reduced growth, although some mutants (e.g., *sch9Δ bwc2Δ*) only showed a phenotype in liquid media; this is likely due to the increased sensitivity of direct competition assay (Supplementary Fig. 1b).

Two major protein kinase signaling pathways emerged from this data set. First, deletion mutants of *YPK1*, *SCH9*, and *GSK3* were hyper-susceptible to elevated $CO_2$ concentrations and all are part of the TOR pathway[11,14], strongly suggesting that TOR is required for $CO_2$ tolerance (Fig. 1b, d). Although Ark1 has not been characterized in *C. neoformans*, its *S. cerevisiae* homolog negatively regulates Tor-dependent endocytosis, tying it to the TOR pathway as well[15]. Second, deletion of the MAPKKK, MAPK, and MAPK kinases (*MKK1*, *BCK1*, and *MPK1*) of the cell wall integrity (CWI) pathway increased $CO_2$ fitness (Fig. 1b, d), suggesting that this pathway may be maladaptive for $CO_2$ stress. Importantly, the TOR and CWI pathways have been shown to negatively regulate the other in both *S. cerevisiae* and *C. neoformans*[14,16].

The only TF mutant that showed reduced fitness (albeit modest) in 5% $CO_2$ was *BWC2*, a component of the blue-white light sensing system in *C. neoformans*[17]. In contrast, multiple TF deletion mutants showed increased fitness in 5% $CO_2$ relative to H99. Among these, a previously defined pathway of Yap1, Gat201 and Gat204[18] emerged (Fig. 1c). Gat201 and Gat204 are GATA TFs that regulate multiple virulence traits in *C. neoformans*[19]. Additionally, Gat201 binds the *GAT204* promoter[19] and regulates its expression of Gat204 (Fig. 1c). Furthermore, Jang et al. found that Yap1 functions upstream of Gat201 during capsule induction[18]. The consistent phenotypes observed across components of previously described regulatory pathways provide confidence that these genes are important for $CO_2$ responses.

Finally, the deletion mutant of Rim101, a TF that is critical for adaptation to host pH[20], is also $CO_2$ resistant (Fig. 1c, e). The cAMP-PKA pathway functions through Rim101 under some conditions[21] and the *pka1Δ* mutant is modestly resistant to $CO_2$ as well (Fig. 1b). From these genetic data, it appears that signaling pathways that are required for adaptation to other host-related environmental stresses such as elevated temperature (CWI pathway) and alkaline pH (Rim101 pathway) may reduce adaptation to host $CO_2$ concentrations. It is also important to note that 9 PKs not related to the TOR pathway with diverse functions are also required for $CO_2$ tolerance (Fig. 1b), further supporting the notion that it represents a significant cellular stress for *C. neoformans*.

### The TOR pathway is required for *C. neoformans* tolerance of elevated $CO_2$ concentrations

A single, essential Tor kinase is present in *C. neoformans* and, consequently, the construction of *TOR1* deletion mutants is not possible[14]. To test the hypothesis that $CO_2$ tolerance is dependent on TOR, we compared the antifungal activity of the TOR inhibitor rapamycin (RAP) under ambient and 5% $CO_2$ conditions. RAP was more potent in 5% $CO_2$ compared to ambient conditions (Fig. 2a). RAP was also more active against an environmental strain, A7-35-23, at elevated $CO_2$ (Fig. 2b). Because deletion mutants of *YPK1* and *SCH9* are both $CO_2$ sensitive and hypersensitive to RAP[14], we asked if resistance to $CO_2$ of TF mutants correlated with RAP resistance. Interestingly, *rim101Δ* mutants were relatively resistant to RAP at elevated $CO_2$ compared to H99 but not under ambient conditions (Fig. 2c). Thus, it is possible that these TF mutants have elevated TOR pathway activity at baseline as a compensatory response to the defects induced by the mutations and this leads to increased fitness in elevated $CO_2$.

The TOR pathway is a conserved regulator of many cellular processes including ribosome biosynthesis, protein translation, amino acid transport, autophagy, actin polarization, and membrane homeostasis among others[22]. Previous studies of the *C. neoformans* TOR pathway confirmed these conserved functions[14]. Specifically, inhibition of TOR with RAP down-regulated genes involved in rRNA

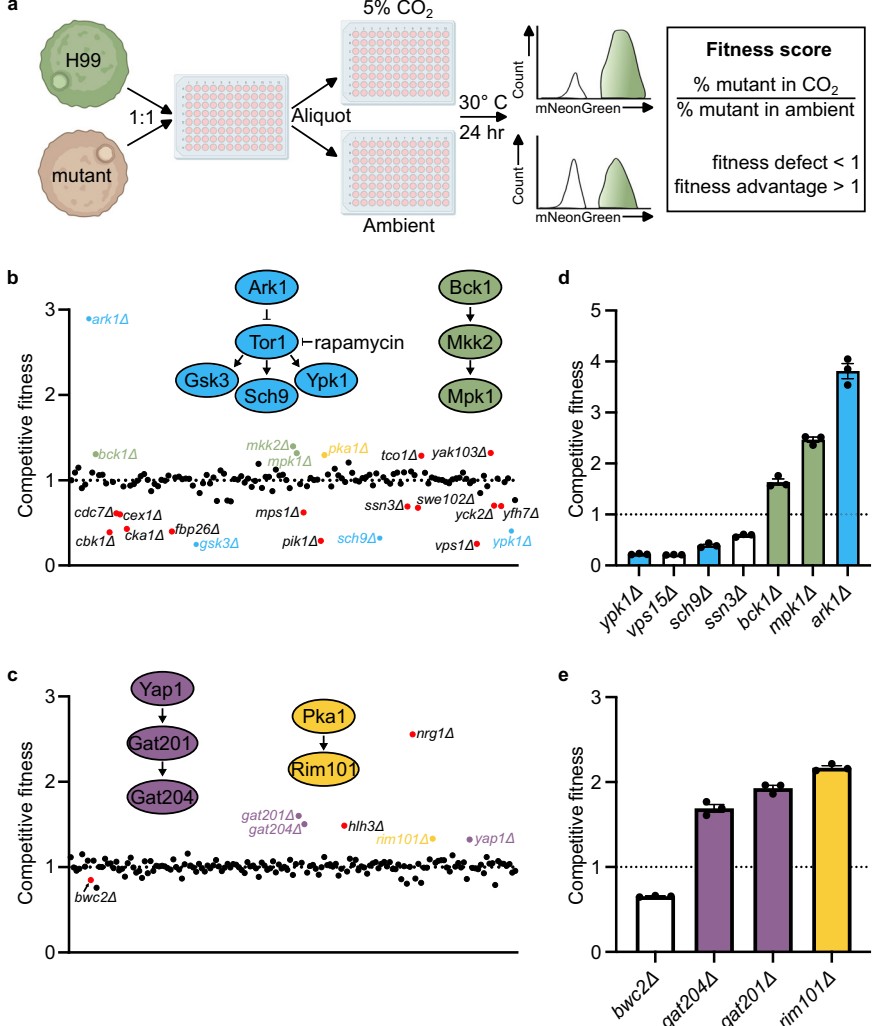

**Fig. 1 | Competition assay reveals involvement of multiple regulatory pathways in response to host levels of $CO_2$.** **a** Schematic representation of competition assay, created with BioRender.com. Overnight cultures of mNeon-Green labeled H99 and unlabeled mutant cells were combined in a 1:1 ratio in RPMI 1640 medium with 165 mM MOPS, pH 7 and incubated at 30 °C for 24 h in ambient air or with 5% $CO_2$. Cell populations were characterized by flow cytometry and the percentage of mNeonGreen negative cells in 5% $CO_2$ normalized to those in ambient conditions to determine a competitive fitness score for each mutant strain, plotted in (**b**) for the kinase deletion library and (**c**) for the transcription factor deletion library. Colored data points indicate values that were statistically significant by CHI square test ($P < 0.05$). Source data and specific $p$-values reported in Supplementary Data 1 and 2. Pathways of interest with multiple significant hits are depicted above each graph with data points involved coordinately colored. Panels (**d**) and (**e**) represent the competitive index of independently generated mutant isolates of strains of interest. Bars represent the average and SEM of three biological replicates. Source data are provided as a Source Data file.

processing, ribosome biogenesis, and actin cytoskeleton while transmembrane transporters and carbohydrate metabolic genes were up-regulated[14]. To further test the hypothesis that $CO_2$ triggers a TOR-mediated cellular response, we characterized the transcriptional profile of H99 cells grown in buffered RPMI medium at 37 °C under ambient or 5% $CO_2$ concentrations (Supplementary Data 3; (Wald test $p$ values and Benjamini-Hochberg adjustment for multiple comparisons)). Somewhat surprisingly, $CO_2$ induced a relatively slow transcriptional response with only 9 genes differentially expressed (adjusted $P$ value < 0.05, $\log_2 \pm 1$) relative to ambient conditions after a 4-hour exposure to 5% $CO_2$ (Fig. 3a).

Eight hours after exposure to 5% $CO_2$, 199 genes were differentially expressed with 192 of those genes downregulated. This set of downregulated genes was enriched for membrane (FDR .012, Benjamini-Hochberg) and integral membrane proteins (FDR 0.024); indeed, 60 of the 199 differentially expressed genes were membrane-associated proteins. By 24 h; however, a total of 1204 genes were differentially expressed ($\log_2 \pm 1$, FDR < 0.05) with the expression of 528

genes downregulated and 677 upregulated (Fig. 3a). GO term analysis (FDR < 0.05, Benjamini-Hochberg) indicates that the upregulated genes are enriched for rRNA processing, ribosome biogenesis, DNA replication and aromatic compound synthesis (Fig. 3b and Supplementary Data 4) while the down-regulated genes were enriched for transmembrane transport, carbohydrate metabolism, redox process, and cellular response to heat (Fig. 3c and Supplementary Data 4). As indicated by the Venn diagram shown in Fig. 3d, 22% of the genes upregulated in $CO_2$ (151/677) are downregulated in cells treated with RAP while 33% (176/528) of genes downregulated in $CO_2$ are upregulated in the presence of RAP. Overall, the transcriptional response to $CO_2$ occurs over multiple hours and appears to have two phases: an early phase that is highly enriched for membrane-associated genes and a late phase that is consistent with the activation of the TOR pathway. The delayed nature of this response would suggest that gene expression may be compensating for the physiological and biophysical effects of elevated $CO_2$ rather than through a direct sensing of the elevated $CO_2$ concentrations.

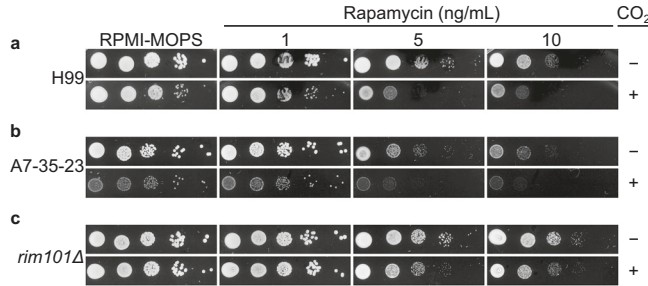

**Fig. 2 | Susceptibility to TOR inhibitor, rapamycin, increases with exposure to host levels of CO₂.** Ten-fold serial dilutions from overnight cultures of (**a**) H99, (**b**) environmental strain A7-35-23 and (**c**) transcription factor mutant *rim101Δ* were spotted on solid RPMI medium with 165 mM MOPS, pH 7 alone or with increasing concentrations of rapamycin as indicated at 30 °C in ambient air or at 5% CO₂ for 48 hours before images were acquired. Images are representative of three biological replicates for each condition.

## Elevated CO₂ suppresses CWI MAPK signaling in CO₂-tolerant but not intolerant strains

Genes involved in the cellular response to heat are downregulated when cells are shifted from ambient to host-like concentrations of CO₂ (Fig. 3c). The CWI pathway is a key positive regulator of the *C. neoformans* response to elevated temperature[23–25]. Because deletion mutants of PKs in the CWI pathway (*BCK1*, *MKK2*, *MPK1*) in H99 background are resistant to CO₂ (Fig. 1b, d), it appears that the expression of temperature stress-related genes may be maladaptive during CO₂ stress. Indeed, we have previously reported that elevated temperature exacerbates CO₂ stress[13]. In *S. cerevisiae* and *C. neoformans*, activation of the TOR pathway negatively regulates the CWI pathway[14,16], suggesting the hypothesis that 5% CO₂ may blunt CWI pathway activation. To test this hypothesis, we used a phospho-specific antibody to monitor the phosphorylation of Mpk1, the terminal MAPK of the CWI pathway, following temperature shift from 30 °C to 37 °C in ambient or 5% CO₂ (Fig. 4a). Consistent with previous literature[23,24], Mpk1 phosphorylation is increased in CO₂-tolerant H99 at 37 °C in ambient air conditions and this level of phosphorylation is maintained for the 4-hour time course. In 5% CO₂, temperature-induced Mpk1 phosphorylation was delayed and was consistently reduced relative to ambient CO₂ conditions over the time course (Fig. 4a, left panel). These data clearly demonstrate that the activation of the CWI pathway is blunted by host levels of CO₂.

We next asked if temperature-induced Mpk1 phosphorylation was also blunted by 5% CO₂ in a CO₂-sensitive strain. To do so, we performed the same temperature-shift experiment with A7-35-23[3,10], a CO₂ sensitive environmental isolate (Fig. 4a, right panel). Mpk1 phosphorylation increased at 37 °C in A7-35-23 but the extent of Mpk1 phosphorylation was unaffected by growth in 5% CO₂. These data, combined with the genetic and transcriptional results discussed above, support a model in which CO₂ suppresses temperature-induced activation of the CWI pathway and that this suppression may improve tolerance to host CO₂ concentrations.

## Reduced Rim101 pathway activity leads to CO₂ and rapamycin resistance

The Rim101 pathway regulates cellular responses to alkaline pH, capsule formation, and cell wall biosynthesis[20,21,26]; our data indicate that it, too, is maladaptive during CO₂ stress (Fig. 1c, e). Rim101 is activated by proteolysis which in turn is mediated by a well-defined protein complex in *C. neoformans* (Fig. 4b). Deletion mutants of two additional components of the Rim101 pathway, the calpain protease Rim13 and the scaffolding protein Rim20, were also resistant to CO₂ relative to H99 (Fig. 4c). Next, we characterized the effect of CO₂

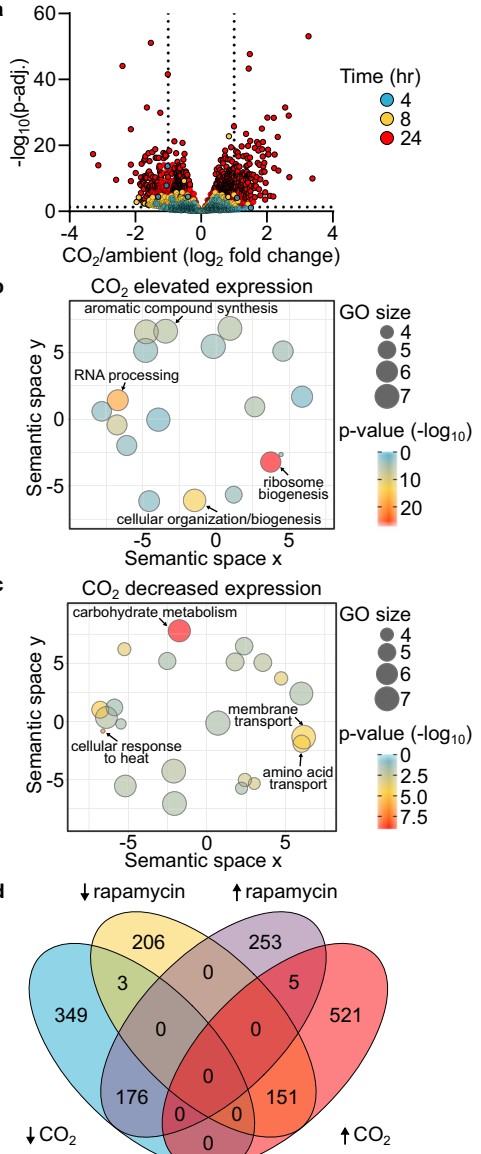

**Fig. 3 | Transcriptional response to carbon dioxide is counter to TOR pathway inhibition. a** Volcano plot of genes identified by RNA-Seq in H99 cultured in RPMI 1640 medium with 165 mM MOPS, pH 7 at 4-, 8- or 24-hours incubation at 37 °C in 5% CO₂ compared to ambient air. Source data and specific *p*-values (Wald test *p* values with Benjamini-Hochberg) are reported in Supplementary Data 3. GO terms for the differentially expressed genes (±log₂ 1 and FDR < 0.05, Benjamini–Hochberg) at 24 h are represented in semantic similarity scatterplots for genes with (**b**) elevated expression or (**c**) decreased expression in 5% CO₂ compared to ambient air. Source data and specific *p*-values reported in Supplementary Data 4. **d** Venn diagram of differentially expressed genes at 24 h in 5% CO₂ compared to ambient air and differentially expressed genes in H99 with 3 ng/mL rapamycin or YPD alone at 3 hours (reported in ref. 14). Source data are provided as a Source Data file.

on the proteolytic processing of Rim101 using western blot analysis of cells containing a Rim101-GFP allele as previously reported[20,21]. Under ambient CO₂ in buffered RPMI medium, both unprocessed (~150 kD band; FL) and processed (~100 kD band; P) Rim101-GFP are detectable (Fig. 4d); additional degradation products are also present in the blot. In 5% CO₂, neither the processed nor the unprocessed forms of Rim101 are detectable and increased amounts of a band corresponding to free GFP are present. These data suggest that Rim101 is degraded in 5% CO₂ and are consistent with the idea that Rim101 activity is reduced to compensate for CO₂ stress.

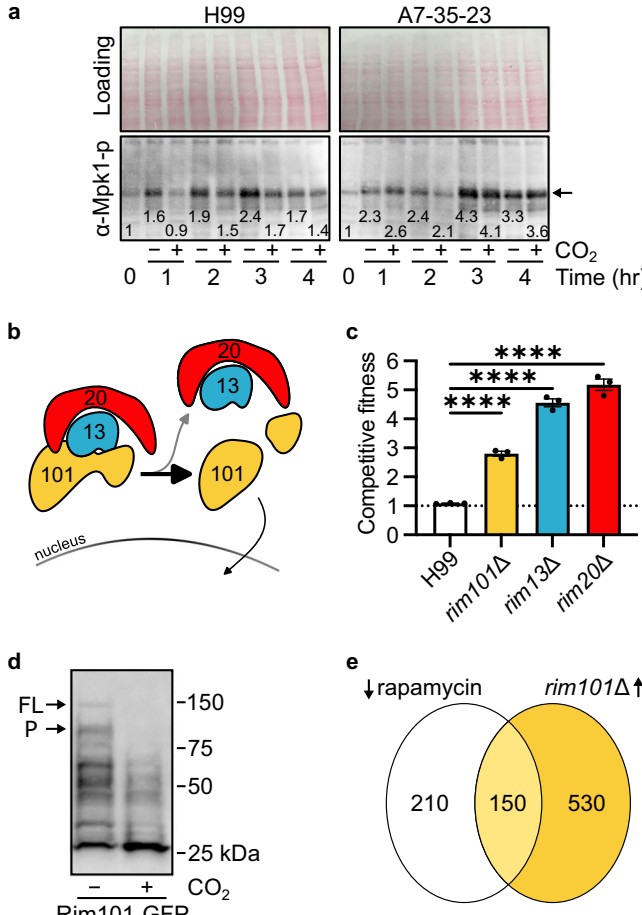

**Fig. 4 | The cell wall integrity pathway and Rim101 activation are maladaptive to carbon dioxide stress. a** Ten µg of total cell lysate from H99 or A7-35-23 cells harvested at 0-, 1-, 2-, 3-, and 4-hours post-shift from 30 °C to 37 °C were analyzed by western blot for phosphorylated-Mpk1 in ambient or 5% $CO_2$ conditions. Cells were grown in YPD, shaking at 200 rpm. Total loading was visualized by Ponceau. Band intensity relative to the 0-hour time point for each strain is indicated. Results are representative of three biological replicates. **b** Schematic representation of Rim101 processing pathway. Cleavage of Rim101 by the Rim13 protease leads to activation and translocation to the nucleus. **c** Overnight cultures of mNeon-Green labeled H99 and unlabeled mutant cells were combined in a 1:1 ratio in RPMI 1640 medium with 165 mM MOPS, pH 7 and incubated at 30 °C for 24 h in ambient air or with 5% $CO_2$. Cell populations were characterized by flow cytometry and the percentage of mNeonGreen negative cells in 5% $CO_2$ normalized to those in ambient conditions to determine a competitive fitness score for each mutant strain as indicated. Bars represent the average and SD from three technical replicates. An ordinary one-way ANOVA with Tukey's multiple comparisons test was performed using GraphPad Prism. Significance relative to H99 is represented; ****$P < 0.0001$; $P_{rim13} = 2.172^{-7}$; $P_{rim20} = 5.014^{-8}$; $P_{rim101} = 5.507^{-5}$. Results are representative of three biological replicates. **d** Overnight cultures of H99 expressing GFP-tagged Rim101 were diluted to an $OD_{600}$:0.2 and grown to an $OD_{600}$:1 in RPMI 1640 with 165 mM MOPS, pH 7 shaking at 200 rpm in ambient air or 5% $CO_2$. Rim101-GFP was pulled down from the whole cell lysate of an equivalent number of cells with GFP-trap beads. An anti-GFP antibody was used to probe the separated proteins on a nitrocellulose membrane. Full-length (FL) and processed (P) Rim101-GFP are indicated. Results are representative of three biological replicates. **e** Venn diagram of genes downregulated by rapamycin treatment in H99 compared to those that are upregulated under $CO_2$ in the *rim101Δ* mutant compared to $CO_2$-exposed H99. RNA-Seq source data and specific p-values (Wald test p values and Benjamini-Hochberg) are reported in Supplementary Data 5. Source data are provided as a Source Data file.

We next characterized the effect of the Rim101 pathway on the transcriptional response to growth at 5% $CO_2$. The *rim101Δ* mutant was incubated in buffered RPMI medium in ambient or 5% $CO_2$ for 24 h. The Rim101 TF has a broad effect on gene expression with 680 genes upregulated and 859 genes downregulated in the *rim101Δ* mutant compared to H99 in 5% $CO_2$ ($\log_2 \pm 1$, FDR < 0.05, Supplementary Data 5; (Wald test p values and Benjamini-Hochberg adjustment for multiple comparisons)). Because the *rim101Δ* mutant is resistant to RAP, we hypothesized that genes upregulated in the mutant may correspond to genes downregulated by RAP. Indeed, 150 of the 360 genes (41%) downregulated by inhibition of TOR[14] are upregulated in the *rim101Δ* mutant (Fig. 4e). GO term analysis of this set demonstrates that ribosome biogenesis (40 genes; $p = 8.3 \times 10^{-37}$), ribonucleoprotein complex processing (40 genes, $p = 1.8 \times 10^{-33}$), and RNA processing (38 genes; $p = 5.8 \times 10^{-22}$) are strongly enriched and represent 25% of the upregulated genes in *rim101Δ*. These genes are key effectors of the TOR pathway and are upregulated beyond the level of expression induced by $CO_2$ in H99. These transcriptional data support the hypothesis that deletion of Rim101 leads to compensatory activation of the TOR pathway and, consequently, increased $CO_2$ tolerance.

## Elevated expression of putative ABC transporter CNAG_07799/ *PDR9* reduces $CO_2$ fitness

Next, we sought to identify non-regulatory genes that modulate $CO_2$ fitness. To do so, we examined the expression of $CO_2$-responsive genes in $CO_2$-tolerant and -intolerant strains using Nanostring nCounter technology. A focused set of 118 genes covering a diverse set of functions that showed differential regulation in $CO_2$ ($\pm 1 \log_2$, FDR < 0.05) based on the RNA seq data at the 24-hour time point above (see Supplementary Data 6 for full set of genes, raw and processed data). The transcriptional profiles of $CO_2$-tolerant and intolerant strains were distinct as shown in Fig. 5a. The sensitive strains show a more consistent profile across different strains than the $CO_2$-tolerant strains. For example, the profile for $CO_2$-tolerant strain C23 is quite distinct from the other two $CO_2$-tolerant strains. In contrast, the $CO_2$-sensitive strains have sets of genes that are consistently highly expressed or lowly expressed. No genes were consistently highly expressed in all tolerant strains relative to intolerant genes. However, a set of 13 genes was expressed higher in the $CO_2$-sensitive stains relative to the tolerant strains (Fig. 5a inset). Five out of these 13 genes have predicted functions related to membrane and lipid homeostasis: *ECM2201*, *CNAG_03227*, *HAPX*, CNAG_07799/*PDR9*, *IPC1*, and *SRE1*. *ECM2201*, *HAPX*, and *SRE1* are TFs with a role in regulating ergosterol biosynthesis genes. Deletion mutants of the three TFs showed $CO_2$ fitness that was similar to WT (Supplementary Data 2). *IPC1* codes for the enzyme that generates inositol-phosphatidyl-ceramide which is the target of the antifungal molecule aureobasidin[27]; increased $CO_2$, however, did not affect the antifungal activity of aureobasidin (Supplementary Fig. 2).

CNAG_07799 is one of 10 PDR/ABCG family ABC transporters in the *C. neoformans* genome and has been named *PDR9* by Winski et al.[28]. Phylogenetic analysis places *PDR9* in Clade III of fungal *PDR/ABC* transporters and its closest *C. neoformans* homologue is *AFR1* which is involved in azole resistance[28]. In addition to mediating drug efflux, PDR/ABC transporters have significant roles in lipid transport and membrane homeostasis[29]. We were, therefore, interested in its potential effect on $CO_2$ tolerance.

The expression of *PDR9* is increased by $CO_2$ exposure in both tolerant and sensitive strains (Fig. 5b). However, the absolute expression of *PDR9* in ambient and 5% $CO_2$ conditions is much higher in sensitive strains. For tolerant strains, the expression of *PDR9* following induction by 5% $CO_2$ remains well below the baseline levels of intolerant strains. These data suggest that it is the expression level of *PDR9*

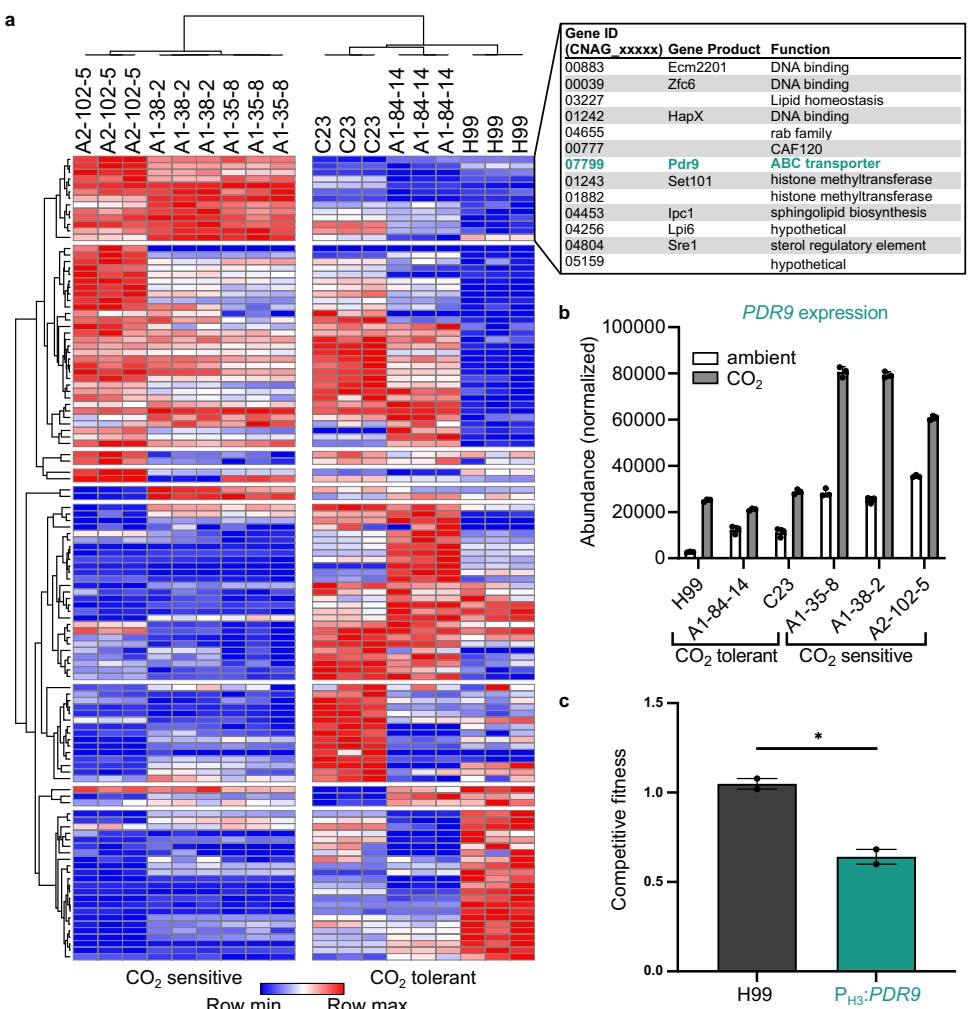

**Fig. 5 | Elevated *PDR9* expression is correlated with CO₂ sensitivity.** Indicated CO₂-sensitive or -tolerant strains were cultured in RPMI 1640 medium with 165 mM MOPS, pH 7 for 24 hours at 37 °C in ambient air or 5% CO₂. Total RNA was isolated from harvested cells, 100 ng of RNA was hybridized to a custom Nanostring probe set and quantified on a Nanostring Sprint nCounter. **a** Normalized counts for strains grown in 5% CO₂ conditions are presented as a heat map hierarchically clustered in Morpheus. Gene names and functions for the subset of 13 genes expressed highly in CO₂-sensitive strains compared to CO₂-tolerant strains are described. Source data are reported in Supplementary Data 6. **b** Normalized Nanostring counts from total RNA for *PDR9* in ambient or 5% CO₂ conditions. **c** Overnight cultures of mNeon-Green labeled H99 and unlabeled H99 or $P_{H3}$:*PDR9* cells were combined in a 1:1 ratio in RPMI 1640 medium with 165 mM MOPS, pH 7 and incubated at 30 °C for 24 h in ambient air or with 5% CO₂. Cell populations were characterized by flow cytometry and the percentage of mNeonGreen negative cells in 5% CO₂ was normalized to those in ambient conditions to determine a competitive fitness score for each strain as indicated. Bars represent the average and SEM from two biological replicates. A two-tailed, unpaired *t* test was performed using GraphPad Prism. *$P < 0.05$; $P = 0.0151$. Source data are provided as a Source Data file.

that is important for CO₂ tolerance and not the fold change from ambient to 5% CO₂. Nanostring profiling of TF mutants with altered CO₂ tolerance also showed that *PDR9* was downregulated in the resistant mutant *rim101Δ* but was upregulated in the hypersensitive mutant *bwc2Δ* (Supplementary Fig. 3a). These data suggested us that elevated expression of *PDR9* may reduce CO₂ fitness.

To test this hypothesis, we constructed a derivative of the CO₂ tolerant strain H99 in which *PDR9* was placed under the control of the Histone 3 promoter which has been used by So et al. to overexpress other genes[14]; importantly, these strains have increased expression of *PDR9* by RT-PCR analysis at both ambient and 5% CO₂ with the expression in CO₂ slightly higher (<1.5 fold, Supplementary Fig. 3b). Consistent with our hypothesis, the $P_{H3}$-*PDR9* strain show reduced fitness in 5% CO₂ relative to its H99 parental strain (Fig. 5c). Similar results were obtained on spot dilution assays (Supplementary Fig. 3c). We also introduced the $P_{H3}$-*PDR9* allele into a second *C. neoformans* background (A1-84-14) and observed an increase in CO₂-susceptibilty, indicating that the phenotype is not dependent on

strain background (Supplementary Fig. 3d). Thus, absolute expression of *PDR9* correlates with reduced growth in host concentrations of CO₂. It is likely that the increased expression of *PDR9* in H99 and other relatively tolerant strains induced by CO₂ contributes to their modest reduction in growth in 5% CO₂ relative to ambient CO₂ conditions (see Fig. 2a and Supplementary Fig. 3c). However, our data indicate that the absolute expression levels and not the difference in expression of *PDR9* between ambient and elevated CO₂ correlate with sensitivity to CO₂.

We previously reported that CO₂ increases *C. neoformans* susceptibility to the ergosterol inhibitor fluconazole and myriocin, an inhibitor of the first step of sphingolipid biosynthesis[10]. Since *S. cerevisiae* homologs of *PDR9* affect susceptibility to these drugs, we tested the susceptibility of the *PDR9* overexpression strain (Supplementary Fig. 3e). There was no difference in the zone of inhibition for fluconazole, indicating that *PDR9* does not act as a fluconazole efflux pump and that it is unlikely that the strains have reduced ergosterol content[28,30]. In contrast, myriocin susceptibility was increased in the

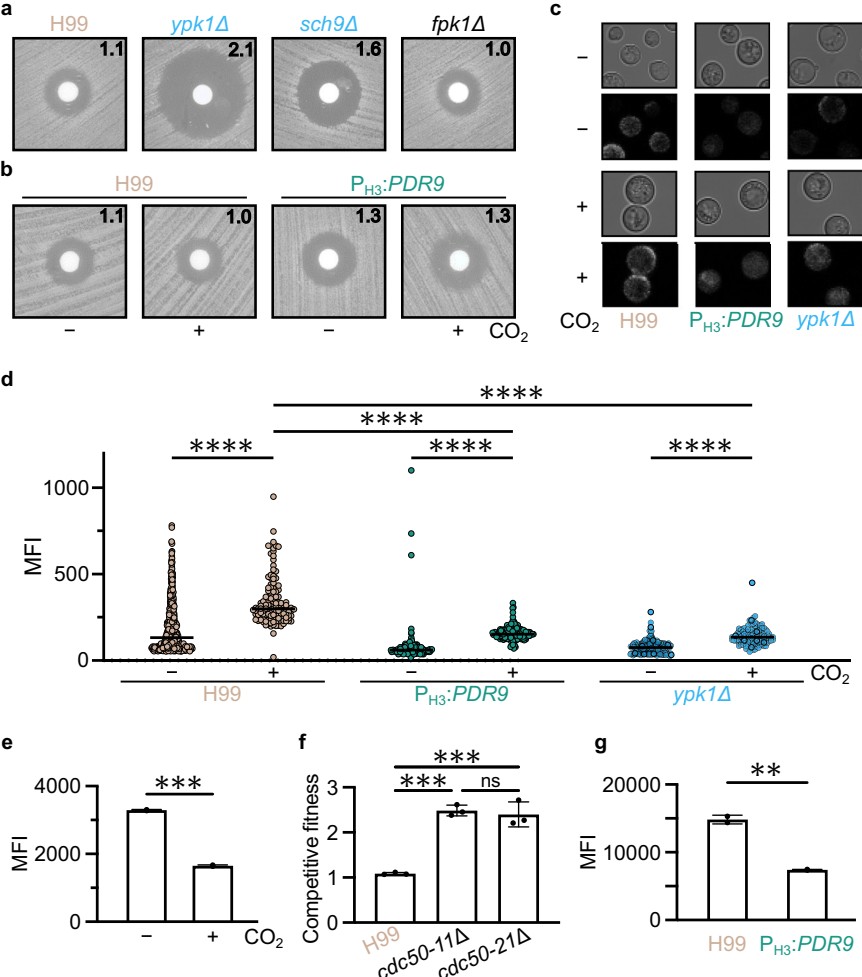

**Fig. 6 | $CO_2$-dependent phospholipid asymmetry is directed through TOR pathway. a**, **b** Cells from overnight cultures of indicated strains were spread on solid RPMI 1640 medium with 165 mM MOPS, pH 7. Sterile disks were placed on plates and 200 μg duramycin was added to each disk. Cells were incubated at 30 °C for 48 h in ambient air or in 5% $CO_2$ before images were acquired. The zone of clearance was measured and indicated in centimeters in the top right corner for each strain/condition. Plates are representative of three biological replicates. **c**, **d** Annexin V staining was performed on cells of indicated strains after 24 h incubation in RPMI 1640 medium with 165 mM MOPS, pH 7 in ambient air or in 5% $CO_2$. Images were captured on a Leica confocal microscope (representative images (**c**)) and mean fluorescence intensity (MFI) was measured in ImageJ software (**d**). At least 100 cells were quantified for each condition. $n_{H99-} = 817$; $n_{H99+} = 128$; $n_{PDR9-} = 263$; $n_{PDR9+} = 206$; $n_{ypk1\Delta-} = 307$; $n_{ypk1\Delta+} = 130$. An ordinary one-way ANOVA was performed using GraphPad Prism with a Sidak follow-up test adjusted for multiple comparisons. ****$P < 0.0001$. $P_{H99-vsH99+} = 1.82$-53; $P_{PDR9-vs.PDR9+} = 3.13^{-18}$; $P_{H99-vs.PDR9+} = 1.15^{-45}$; $P_{H99+vsPDR9+} = 4.18^{-50}$; $P_{ypk1\Delta-vs.ypk1\Delta+} = 1.29^{-9}$; $P_{H99-vs.ypk1\Delta-} = 1.75^{-48}$; $P_{H99+vs.ypk1\Delta+} = 9.25^{-49}$. **e**, **f** Cells from overnight cultures of

H99 were cultured in RPMI 1640 medium with 165 mM MOPS, pH 7 at 30 °C in ambient or 5% $CO_2$ for 18 h to mid-log phase. Cells were washed and labeled with NBD-phosphatidylserine (**e**) or NBD-phosphatidylethanolamine (**f**) for 30 min before washing and assessment of fluorescence by flow cytometry. Bars represent MFI with SEM from two biological replicates. Graphs are representative of experiments performed on three independent days. A two-sided, unpaired *t* test was performed using GraphPad Prism. ***$P < 0.001$. **e** $P = 0.0002$ (**f**) $P = 0.0038$. **g** Overnight cultures of mNeon-Green labeled H99 and unlabeled H99 or $cdc50\Delta$ cells were combined in a 1:1 ratio in RPMI 1640 medium with 165 mM MOPS, pH 7 and incubated at 30 °C for 24 h in ambient air or with 5% $CO_2$. Cell populations were characterized by flow cytometry and the percentage of mNeonGreen negative cells in 5% $CO_2$ was normalized to those in ambient conditions to determine a competitive fitness score for each strain as indicated. Bars represent the average and SEM from three biological replicates. An ordinary one-way ANOVA with Tukey's multiple comparisons test was performed using GraphPad Prism. ***$P < 0.001$. $P_{H99 vs. cdc50-11\Delta} = 0.0002$; $P_{H99vs. cdc50-21\Delta} = 0.0002$. Source data are provided as a Source Data file.

overexpression strain, suggesting that increased expression of *PDR9* may alter sphingolipid homeostasis.

## *C. neoformans* remodels phospholipid asymmetry through the TOR pathway in response to $CO_2$ stress

The TOR pathway regulates phospholipid asymmetry at the plasma membrane (PM) by modulating the activity of aminoglycerolipid (AGL) flippases via Ypk1 in *S. cerevisiae*[31]. The flippases transport phosphatidylethanolamine (PE) and/or phosphatidylserine (PS) from the outer leaflet of the PM to the inner, cytosolic face of the PM and floppases transport PE/PS in the opposite direction[32]. Based on the phenotypes of TOR mutants and their function in *S. cerevisiae*, we hypothesized

that PM asymmetry may play an important role in *C. neoformans* tolerance of host $CO_2$ stress.

To test this hypothesis, we first examined the effect of $CO_2$ and mutants with altered $CO_2$ tolerance on susceptibility to the antifungal duramycin. Duramycin binds PE on the PM outer leaflet as part of its mechanism of action and cells with increased outer leaflet PE exposure show increased susceptibility to the drug[31]. Although the susceptibility of H99 to duramycin is not affected by 5% $CO_2$, the *ypk1Δ* mutant is sensitive to duramycin, relative to H99 (Fig. 6a, b). In *S. cerevisiae*, the TOR pathway inhibits PE flippases though activation of Ypk1 which, in turn, is a negative regulator of Fpk1; accordingly, overexpression of *YPK1* increases duramycin susceptibility while the *fpk1Δ* deletion

mutant is resistant to duramycin[31]. These data suggest that Ypk1 has a distinct role in PE homeostasis in *C. neoformans*. Supporting this conclusion, the duramycin susceptibility of the *fpk1Δ* mutant is unchanged relative to H99 (Fig. 6a). Interestingly, deletion of *SCH9*, a second TOR dependent kinase that is also hypersusceptible to $CO_2$, has increased susceptibility to duramycin. If Ypk1 and Fpk1 have the same lipid asymmetry functions in *C. neoformans* and *S. cerevisiae*, then the *ypk1Δ* mutant would be resistant to duramycin and the *fpk1Δ* mutant would be hypersusceptible. These results indicate that the Tor-Ypk1 axis has a completely different effect on PE asymmetry in *C. neoformans* compared to *S. cerevisiae*. Ypk1 is required for $CO_2$ tolerance while deletion of *FPK1* has no effect on this phenotype (Supplementary Data 1), further supporting the idea that the relationship between these two TOR-regulated kinases is different in *C. neoformans*

One of the proposed roles of PDR/ABC transporters in lipid homeostasis is as floppases[29]. Because TOR-related mutants with increased PM outer leaflet PE exposure show increased $CO_2$ susceptibility, we tested the duramycin susceptibility of the $P_{H3}$-*PDR9* strain. As shown in Fig. 6b, overexpression of *PDR9* increases susceptibility to duramycin at both ambient and 5% $CO_2$. The changes in zones of clearance observed with the $P_{H3}$-*PDR9* strain are similar to those observed in *S. cerevisiae* mutants involved in PE homeostasis[31]. These data are consistent with *PDR9* having a possible PE floppase activity and that its effect on $CO_2$ tolerance may involve a role in membrane lipid homeostasis. The definitive biochemical characterization of floppases has been technically difficult and somewhat controversial. In *S. cerevisiae*, Pdr5 has been proposed to be a floppase[33]. Pdr5 is primarily localized to the PM[33]. We, therefore, tagged *PDR9* with mNeonGreen under both the endogenous and $P_{H3}$ promoter; the latter strain was hypersusceptible to $CO_2$, confirming the mNeonGreen-tagged allele is functional (Supplementary Fig. 4a). In both strains and under multiple growth conditions, Pdr9-mNeonGreen localizes to intracellular puncta and not to the plasma membrane (Supplementary Fig. 4b). Since validated organelle markers in *C. neoformans* are limited, we have not conclusively localized Pdr9-mNeonGreen, but the pattern of signal is characteristic of late-Golgi/endosomes and not the PM. Thus, Pdr9 affects PE homeostasis in a manner consistent with a floppase. However, Pdr9 does not localize to the PM, which is where most proposed floppases are found, raising the possibility that this effect on PE distribution may be indirect. As such, additional biochemical and cell biological studies will be required before a definitive lipid asymmetry-related function for Pdr9 can be made with confidence.

Flippases and floppases also affect the distribution of PS between the inner and outer leaflets of the PM[32]. To test the effect of $CO_2$ on the distribution of PS, we used the cell impermeant molecule, annexin V, to stain outer leaflet PS following recently procedures reported by Huang et al. for the use of this assay with *C. neoformans*[34]. Consistent with the previous report[34], outer leaflet PS is low but detectable under ambient air conditions (Fig. 6c, d). Exposure to 5% $CO_2$ in RPMI at 37 °C significantly increases outer leaflet PS staining in H99. Overexpression of *PDR9* reduced the level of outer leaflet PS induced by 5% $CO_2$ as did deletion of *YPK1*. Together, these data clearly indicate that adaptation to 5% $CO_2$ is associated with increased outer membrane PS relative to PE and that disruption of this balance by increased expression of *PDR9* or by deletion of *YPK1* leads to reduced fitness in 5% $CO_2$.

To further test the hypothesis that elevated $CO_2$ concentrations increase PM outer leaflet PS concentrations by altering phospholipid asymmetry, we performed PS uptake assays using fluorescently labeled NBD-PS following a previously reported approach with some modifications[34]. Mid-log phase H99 cells grown in ambient conditions or 5% $CO_2$ were exposed to NBD-PS for 30 min and cellular uptake of NBD-PS quantitated by flow cytometry; intracellular uptake of NBD-PS was confirmed by microscopy (Supplementary Fig. 5a). As shown in Fig. 6e, cells exposed to 5% $CO_2$ show ~50% reduction NBD-PS uptake

relative to cells grown in ambient air conditions. Cdc50 regulates the activity of flippases and, consequently, PS asymmetry in *S. cerevisiae*. Multiple labs have shown that Cdc50 is also required for PS flip activity in *C. neoformans*[34,35]. We, therefore, tested the effect of a *cdc50Δ* mutant on $CO_2$ susceptibility. Consistent with previous data showing that the *cdc50Δ* mutant has increased annexin V staining, it is resistant to 5% $CO_2$ relative to WT (Fig. 6f).

An alternative possibility is that total cellular PS is increased in the presence of 5% $CO_2$ leading to the apparent increase in outer leaflet PS localization. To test that possibility, we modulated the total cellular synthesis of PS using a strain containing a copper-repressible allele of *CHO1*, an enzyme required for the synthesis of PS[36]. If $CO_2$ tolerance is dependent on an increase in total PS, then preventing that increase by incubating the $P_{CTR4}$-*CHO1* strain on media containing high copper concentrations should reduce $CO_2$ tolerance. As expected, the $P_{CTR4}$-*CHO1* strain has a growth defect on copper-replete media compared to copper-deficient media at ambient air (Supplementary Fig. 6); however, there is no further reduction in the growth of the strain when it is incubated at 5% $CO_2$. The results of this experiment are more consistent with $CO_2$ tolerance being dependent on lipid asymmetry remodeling than with a mechanism involving a global increase in PS synthesis.

Finally, increased expression of *PDR9* leads to increased duramycin sensitivity which indicates an increase in outer membrane PE (Fig. 6b). To test if this phenotype is associated with alterations in PE transport, we compared the uptake of fluorescently labeled PE as well; intracellular uptake of NBD-PE was also confirmed by microscopy (Supplementary Fig. 5b). As shown in Fig. 6g, overexpression of *PDR9* leads to a 2.5-fold reduction in uptake of NBD-PE. These data further support, but do not conclusively establish, the possible function of Pdr9 as a phospholipid floppase. Taken together, these data strongly support a model in which host-relevant concentrations of $CO_2$ lead to remodeling of PS lipid asymmetry and that disruption of this remodeling reduces tolerance of those conditions.

## Overexpression of *PDR9* reduces *C. neoformans* virulence

Environmental strains with elevated expression of *PDR9* are not only intolerant of host-relevant concentrations of $CO_2$ but are also less virulent in mouse models of cryptococcosis[3,10]. We, therefore, hypothesized that the reduced $CO_2$ tolerance of the $P_{H3}$-*PDR9* strain would translate to reduced infectivity and virulence. To test this hypothesis, we used two infection models. First, we carried out a competitive fitness experiment in which a 1:1 mixture of H99 and $P_{H3}$-*PDR9* was used to infect the respiratory tract of mice. The mice were euthanized at DPI 14 and a ratio of the two strains in the lung tissue was determined by quantitative plating on YPD and YPD + NAT (selective for the overexpression strain). Consistent with in vitro studies, the *PDR9* overexpression strain was 10-fold less fit in the lung (Fig. 7a). We also compared the virulence of H99 and the $P_{H3}$-*PDR9* strain using single-strain infection experiments. As shown in the survival curve in Fig. 7b, infection with H99 led to 70% moribundity by DPI 40 while no animals infected with the $P_{H3}$-*PDR9* strain showed signs of distress to day 86. The fungal burden at day of sacrifice was determined for five H99-infected animals and compared to the fungal burden of $P_{H3}$-*PDR9* strain-infected animals at DPI 86. Consistent with expectations, H99-infected animals showed high fungal burden in both the lung (Fig. 7c) and brain tissue (Fig. 7d). Interestingly, 4 out of five animals infected with the $P_{H3}$-*PDR9* strain had ~3 $log_{10}$ organisms in the lung but none in the brain. This observation suggests that overexpression of *PDR9* reduces lung burden and reduces dissemination to the brain.

To determine if the reduced virulence of the $P_{H3}$-*PDR9* strain was due to alterations in the three canonical *C. neoformans* virulence traits[2], we compared its growth at 37 °C, melaninization, and capsule formation to H99 under both ambient and 5% $CO_2$ conditions. The $P_{H3}$-*PDR9* strain grew similar to H99 at 37 °C (Supplementary Fig. 7a) and

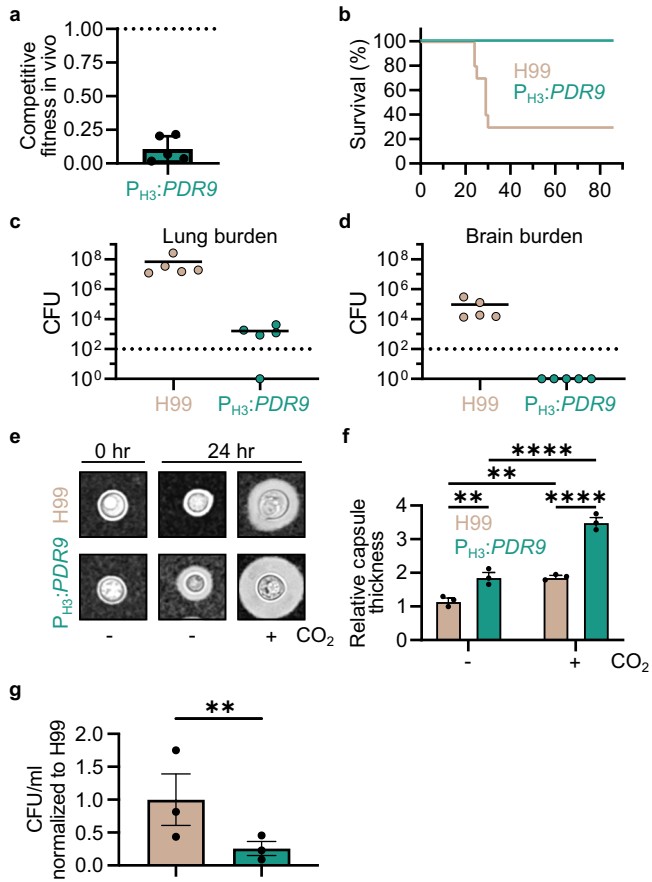

**Fig. 7 | CO₂ regulated expression of *PDR9* is required for virulence. a** Five CD1 mice were infected intranasally with a 1:1 ratio of H99 and P$_{H3}$:*PDR9* cells (5 × 10⁴ per mouse). At DPI 14, mice were sacrificed and the lung was dissected. For fungal burden quantifications, lungs were homogenized and serially diluted, then plated onto YNB and YNB with 100 μg/mL of nourseothricin (NAT) and incubated at 30 °C for two days before counting CFUs. The competitive fitness in vivo was determined by dividing the CFU of P$_{H3}$:*PDR9* cells (NAT resistant) by H99 cells. Data are represented with the average and SD. For the survival study (**b**), ten CD1 mice per group were infected intranasally with 1 × 10⁴ fungal cells. Fungal burden was examined for five mice at the time of termination or at DPI 86. For fungal burden quantifications, lungs (**c**) or brains (**d**) were processed as above. **e, f** H99 and P$_{H3}$:*PDR9* cells were grown in RPMI 1640 + 165 mM MOPS for 24 h at 37 °C in ambient air or 5% CO₂, then prepared for microscopy by counterstaining with India ink (representative images (**e**)). At least 50 cells were quantified per biological replicate, with three biological replicates per condition and processed in ImageJ software to measure capsule. Average capsule thickness at 24 h was normalized to 0 h for each biological replicate (**f**). Bars represent average and SEM. Two-way ANOVA with Tukey's multiple comparisons test was performed in GraphPad Prism. **\*\****P* < 0.01, \*\*\*\**P* < 0.0001. *P*$_{H99\text{-vs.PH3:PDR9-}}$ = 0.0073; *P*$_{H99\text{-vs.H99+}}$ = 0.0072; *P*$_{PH3:PDR9\text{-vs.PH3:PDR9+}}$ = 2.49⁻⁵; *P*$_{H99+\text{vs.PH3:PDR9+}}$ = 2.51⁻⁵. **g** J774 murine macrophage cells were incubated with opsonized H99 or P$_{H3}$:*PDR9* cells. Phagocytosis was assessed after 3 h by washing away non-phagocytosed yeast cells, lysing macrophages, and plating on solid YPD. Plates were incubated at 30 °C for 2 days before counting CFUs. CFU/ml were normalized to H99. Bars represent the average and SEM from three experimental replicates. A two-sided, ratio paired t test was performed using GraphPad Prism. **\*\****P* < 0.01, *P* = 0.0044. Source data are provided as a Source Data file.

showed no changes in melaninization (Supplementary Fig. 7b). Unexpectedly, the capsule formed by the P$_{H3}$-*PDR9* strain was 1.5-fold thicker relative to H99 in 5% CO₂ (Fig. 7e, f). In mouse models, reduced capsule formation is generally associated with reduced virulence. One possible explanation for this discordance is that the capsule formed by P$_{H3}$-*PDR9* is quantitatively larger but may have qualitative or structural defects that affect its function. One of the best characterized functions

of a capsule is to interfere with phagocytosis by macrophages[37]. Using the murine macrophage cell line J774, we compared the phagocytosis of the P$_{H3}$-*PDR9* strain to H99. Consistent with its increased capsule formation, phagocytosis of the P$_{H3}$-*PDR9* strain was reduced relative to H99 (Fig. 7g). This suggests that, to a first approximation, the capsule formed by this mutant functions as expected. Thus, we assert that the best explanation for the reduced fitness of the P$_{H3}$-*PDR9* strain in vivo is its reduced ability to tolerate host levels of CO₂.

## Discussion
Our group recently reported that the ability of *C. neoformans* to tolerate host levels of CO₂ correlates with virulence in strains that express the three most well-characterized virulence traits[10]. Here, we identify the TOR pathway and remodeling of plasma membrane lipid asymmetry as key factors contributing to the ability of *C. neoformans* to tolerate host CO₂ stress. The role of these processes as well as other PKs involved in stress responses support the conclusion that the transition from ambient CO₂ concentrations in the environment to the 100-fold higher levels in the host represents an independent stress in *C. neoformans*; further supporting this conclusion is the large number of genes that are differentially expressed in cells replicating in host concentrations of CO₂ compared to ambient conditions. Of the 15 mutants with reduced fitness in host CO₂ conditions in vitro, twelve have reduced fitness in vivo as well, strongly supporting the concept that CO₂ tolerance/adaptation is required for *C. neoformans* virulence[10].

At the same time, two pathways that regulate *C. neoformans* adaptation to other features of the host environment, the CWI and Rim101 pathways, suppress CO₂ tolerance. In addition, a transcription factor circuit that induces the expression of genes required for virulence, the Yap1-Gat201-Gat204 pathway, is also relatively maladaptive in the setting of CO₂ stress. Host CO₂ stress, therefore, appears to present *C. neoformans* with a conundrum in which pathways such as TOR must be activated for it to replicate while at the same time other pathways required for host adaptation must be suppressed. Very few *Cryptococcus* strains can adapt to mammalian physiology[3,4] and we propose that the requirement for the fungus to balance contradictory physiological responses to host CO₂, pH, and temperature may contribute to this bottleneck.

Our previous work has demonstrated that host temperature (37 °C) exacerbates CO₂ stress[13] and, therefore, it would be reasonable to expect the CWI pathway to be important for tolerance of CO₂. Instead, CO₂ suppresses activation of the CWI pathway, likely through the activation of the TOR pathway. There are, however, multiple signaling pathways that are required for temperature tolerance. We propose that suppression of CWI activity necessitates activation of the other pathways to mediate critical compensatory responses during adaptation to both the CO₂ and temperature stresses of the host environment. Supporting this model is the fact that the RAM pathway is required for *C. neoformans* adaptation to host CO₂ concentrations and temperature[13]. Furthermore, 10/14 PK mutants with reduced fitness in host CO₂ are also hypersensitive to elevated temperature[11]. Therefore, our data suggest that CO₂ and temperature are interrelated host stressors that require responses that must be finely balanced for *C. neoformans* to establish infection.

Although the TOR pathway plays pleiotropic roles in eukaryotic biology, we are unaware of previous reports linking it to CO₂ responses. In fungi, however, studies of the biology of CO₂ have focused almost entirely on the mechanisms by which these organisms generate sufficient bicarbonate in low CO₂ environments such as the external air[38]. Two of the most well-studied functions of the TOR pathway are translational regulation and phospholipid homeostasis[14,31]. Our results indicate that both of these functions are critical for *C. neoformans* adaptation to host CO₂ concentrations. The induction of canonical TOR target genes such as ribosomal

components and translational machinery strongly supports this conclusion. Additional studies, however, will be needed to identify the specific transcription factors that mediate this transcriptional response because our screen of single gene deletion mutants has not identified candidate regulators. This suggests that there are likely to be overlapping TFs with redundant functions and that genetic interaction analysis will be needed to uncover those that are critical to the transcriptional response to $CO_2$.

Our work has shed light on the role of the TOR pathway in maintaining lipid homeostasis at host $CO_2$ concentrations in *C. neoformans*. Previous studies in *S. cerevisiae*[39] as well as in *C. neoformans*[40] have shown that TOR acts through its target Ypk1 to regulate both phospholipid asymmetry at the plasma membrane and sphingolipid biosynthesis. The *ypk1Δ* mutant is highly sensitive to $CO_2$, indicating that this arm of the TOR pathway is also critical to high $CO_2$ adaptation. We previously reported that $CO_2$ increases *C. neoformans* susceptibility to the sphingolipid biosynthesis inhibitor myriocin[10]. The Del Poeta lab has also found that mutants (*smt1Δ* and *gcs1Δ*) involved in sphingolipid biosynthesis have specific growth defects in host-like in vitro conditions containing 5% $CO_2$[41]. Taken together, these observations are consistent with the conclusion that TOR-Ypk1-regulated sphingolipid biosynthesis is an important cellular response to $CO_2$ in *C. neoformans*.

In *S. cerevisiae*, TOR-Ypk1 regulates the function of phospholipid flippases responsible for maintaining the asymmetric distribution of PS and PE at the plasma membrane[31,39]. Our data suggests that the TOR-Ypk1 axis functions differently in *C. neoformans* compared to *S. cerevisiae*. The *C. neoformans ypk1Δ* mutant is sensitive to duramycin, an antifungal molecule that binds outer leaflet PE. The same mutant in *S. cerevisiae* is as sensitive to duramycin as WT while overexpression of *YPK1* increases sensitivity[31]. In *S. cerevisiae*, Ypk1 inhibits the function of Fpk1 which activates flippases[31]. Consequently, deletion of *FPK1* increases duramycin sensitivity in *S. cerevisiae*[31]; in *C. neoformans*, deletion of *FPK1* has no effect on duramycin susceptibility or on *C. neoformans* $CO_2$ tolerance. Thus, it appears that the function of the Ypk1-Fpk1 pair is distinct in *C. neoformans* compared to *S. cerevisiae*. Fpk1 contains a sequence that corresponds to the canonical Ypk1 substrate motif, suggesting it may still be a substrate for Ypk1[42]. However, Bahn and co-workers, as part of their large-scale analysis of PKs in *C. neoformans*[11], also observed phenotypes that were inconsistent with the conservation of the Ypk1-Fpk1 relationship between *S. cerevisiae* and *C. neoformans*. Additional work will be needed to understand the molecular mechanisms underlying these apparent changes in Tor-Ypk1-Fpk1 regulation of PE homeostasis.

The TOR-Ypk1 axis has also been implicated in the regulation of PS distribution at the plasma membrane in *S. cerevisiae*[43]. Whereas there appears to be little change in PE distribution in response to $CO_2$ (Fig. 6b), PM outer leaflet PS exposure increases in H99 cells in host-relevant $CO_2$ concentrations in a Ypk1-dependent manner (Fig. 6d). In the case of PS homeostasis, therefore, Ypk1 is functioning similarly in both *C. neoformans* and *S. cerevisiae*. It is somewhat surprising that the *C. neoformans* responds to $CO_2$ by increasing outer membrane PS because this is most commonly an indication of apoptosis in eukaryotic cells[32]. It is important to note that 5% $CO_2$ does not kill either tolerant or intolerant *C. neoformans* strains but is fungistatic[10]; therefore, increased PS exposure does not appear to be an indication of reduced viability. At this point, it is unclear how increased PM outer leaflet PS suppresses the fungistatic effects of elevated $CO_2$. Our transcriptional and phenotypic data strongly support the notion that elevated $CO_2$ concentrations affect plasma membrane and lipid homeostasis. In general, PS supports membrane fluidity whereas increased $CO_2$ concentrations are reported to decrease the fluidity of microbial membranes[44]. Based on these opposing biophysical effects of PS and $CO_2$ on biological membranes, one potential explanation for the increased $CO_2$ fitness of *C. neoformans* cells is that the increased outer leaflet PS promotes membrane fluidity to counter the membrane rigidifying effects of elevated $CO_2$.

Our data also indicate that the increased amount of PM outer leaflet PS in *C. neoformans* at host concentrations of $CO_2$ is due to reduced uptake of PS and not due to a global increase in cellular PS. Previous studies have confirmed that Cdc50[34,35], a conserved flippase subunit that regulates multiple flippases, is required for PS flippase function in *C. neoformans*. Consistent with that function, deletion of *CDC50* increases $CO_2$ tolerance, further supporting the conclusion that increased PM outer leaflet PS reduces the toxicity of host concentrations of $CO_2$. Another possible mechanism for increased PM outer leaflet PS levels, would be an increase in transport from the inner leaflet to the outer leaflet also known as PS floppase activity. To date, no bona fide PS floppase has been described in fungi although Pdr5 appears to function as a PE floppase in *S. cerevisiae*[33]. Still, many ABC/PDR genes remain to be characterized and, in principle, could carry out this function[28]. Overall, our data and prior literature precedents are most consistent with a mechanism in which $CO_2$ triggers a cellular response that dramatically reduces PS uptake and leads to a corresponding increase in outer leaflet PS.

Although annexin V staining is widely used as an assay for alterations in PS distribution between the inner and outer leaflets of the plasma membrane, some studies have indicated that annexin V is not specific for PS and can also bind to other anionic phospholipids such as PE, phosphatidylglycerol and phosphatidic acid[45,46]. Our uptake assays indicate that there is decreased transport of PS into the cells in the presence of $CO_2$, supporting the notion that PS distribution is altered. However, it is also possible that the increased Annexin V staining observed in elevated $CO_2$ concentrations is due to the accumulation of other anionic phospholipids on the outer leaflet of the plasma membrane. Since duramycin sensitivity does not change at elevated $CO_2$, other anionic phospholipids would most likely be responsible for these effects. In either case, our data strongly support the notion that tolerance of host concentrations of $CO_2$ requires a significant remodeling of outer membrane anionic phospholipid distribution; for simplicity, we will refer to PS in the remainder of the discussion with the understanding that other anionic phospholipids may also be involved.

The increased PM outer leaflet PS induced by host $CO_2$ levels may also be linked to the increased fitness of RIM101 pathway mutants under those conditions. Brown et al. found that under neutral pH the *cdc50Δ* mutant has delayed proteolytic processing of Rim101[47]. This result indicates that increased PM outer leaflet PS (or reduced PM inner leaflet PS) negatively regulates Rim101 pathway activation. Our results suggest that both increased PM outer leaflet PS and reduced Rim101 pathway activity are involved in the adaptation of *C. neoformans* to host $CO_2$ conditions. Based on the findings of Brown et al.[47], these two observations may be mechanistically related.

Finally, our initial characterization of Pdr9 suggests that it is not a typical fungal ABC/PDR transporter. Overall, Pdr9 clearly affects plasma membrane lipid homeostasis based on the following $P_{H3}$-*PDR9* strain phenotypes: (1) increased susceptibility to the PE-targeted antifungal duramycin; (2) increased susceptibility to the sphingolipid biosynthesis inhibitor myriocin; (3) suppression of $CO_2$-induced PM outer leaflet localized PS; and (4) reduced uptake of PE. As such, Pdr9 shows features of a PE floppase (phenotypes 1 and 4) and a PS flippase (phenotype 3). To date, all experimentally characterized flippases are P4-ATPases[48]. Therefore, the effect of Pdr9 on PS asymmetry is far more likely to be indirect rather than through a direct PS flippase activity.

ABC family transporters are proposed to function as lipid floppases[32,33]. The best characterized examples are ABCA group (not ABCG/PDR family) mammalian transporters of phosphatidylcholine and sphingomyelin (e.g., ABCA1[49,50]). Although the ABCG/PDR group *S. cerevisiae* Pdr5 has been proposed to be a PE floppase[32,33] and our

Pdr9 data are consistent with that function, it also affects sphingolipid biosynthesis which could contribute to its effect on PS and $CO_2$ phenotypes. Taken together, the complex effect of elevated expression of *PDR9* on lipid asymmetry suggests it does so through a combination of direct and indirect effects on lipid homeostasis.

Supporting the complex role that Pdr9 plays in lipid asymmetry and distinguishing it from a typical fungal ABC/PDR transporter is its apparent lack of PM localization. Instead, it localizes to intracellular puncta that show features of the late Golgi or endosome compartment. ABC transporters involved in lipid homeostasis have been localized to these compartments in mammalian systems[51], providing some precedent for the correlation between the function and localization of Pdr9. A wide range of additional cell biological, biochemical and biophysical studies will be required to establish the molecular function of Pdr9. Regardless of its likely indirect and complex mechanism of action toward $CO_2$ tolerance, the effect of *PDR9* on multiple facets of lipid homeostasis and $CO_2$ susceptibility has provided important insights into the mechanisms of the latter.

In summary, the ability of *C. neoformans* to tolerate host concentrations of $CO_2$ is a trait that distinguishes low virulence strains from high virulence strains. Here, we have found a novel function of the TOR pathway as a critical mediator of adaptive $CO_2$ responses in the $CO_2$-tolerant strain H99. We also show that TOR-Ypk1 mediated remodeling of plasma membrane lipid asymmetry toward increased PS exposure is associated with $CO_2$ tolerance.

## Methods

### Ethics statement
This study was performed according to the guidelines of the NIH and the University of Georgia Institutional Animal Care and Use Committee (IACUC). The animal models and procedures used have been approved by the IACUC (AUP protocol number: A2020 06-015).

### Strains and growth conditions
Yeast extract-peptone-2% dextrose (YPD) and synthetic complete (SC) were prepared according to standard recipes[52]. RPMI 1640 without glutamine or sodium bicarbonate was buffered with 165 mM MOPS and pH adjusted to 7. Strains used in this work are described in Supplementary Data 7. *C. neoformans* kinase and transcription factor deletion libraries were acquired from the Fungal Genetic Stock Center (FGSC). H99 reference strain, environmental and clinical *C. neoformans* strains were generous gifts from A. Litvintseva, T. Mitchell, and J. Perfect. The *rim13Δ*, *rim20Δ* and *rim101*-GFP strains were generous gifts from J.A. Alspaugh. $P_{CTR4}$:*CHO1* strain was a generous gift from C. Xue. The *cdc50Δ* strains were a generous gift from J. Kronstad. Strains were stored at −80 °C in 20% glycerol. Frozen stocks were recovered on solid YPD medium at 30 °C for 2 days. To prepare for assays, 3 mL YPD liquid was inoculated per strain and grown overnight, 30 °C, shaking at 200 rpm.

### Strain construction
For all strains generated in this work (Supplementary Data 7), CRISPR/Cas9 short-arm homology (SAH) and transient CRISPR-Cas9 coupled with electroporation (TRACE) methods were used as published[53,54]; CRISPR components were PCR purified and transformed into H99 via electroporation using the "Pic" setting on a Bio-Rad Micropulser.

For mNeonGreen-expressing H99, an intact pGWKS11 (a gift from James Fraser (Addgene plasmid #139418; http://n2t.net/addgene: 139418; RRID: Addgene_139418[55])) was transformed with Cas9 DNA and two sgRNAs targeting the Safe Haven 1 (SH1) location (generated with primers LCR075-LCR078, (oligonucleotide sequences in Supplementary Data 8)) into H99. Transformants were selected on YPD medium with 100 µg/mL nourseothricin (NAT) and fluorescent isolates were identified by flow cytometry. A single isolate was selected for use

in this work by equivalent competition with wild-type H99 over 48 h at 30 °C and 37 °C in ambient or 5% $CO_2$ conditions.

For gene deletion constructs, SAH Repair 5′ and 3′ oligos designed with a 50 bp sequence matching the flanks of the H99 target gene sequence and a 20 bp sequence matching the flanks of the NAT or hygromycin (HygB) resistance marker cassette were used to amplify a repair construct, which replaced the entire open reading frame for the target gene with the NAT or HygB cassette. For deletion of *ssn3*, *vps15* and *ypk1*, primers L1 and R2 from the kinase deletion knockout collection[11] were paired to amplify the resistance cassette from knockout specific genomic DNA and used as a repair construct. Repair constructs were transformed with Cas9 DNA and two gene-specific sgRNAs into H99. Transformants were selected on YPD medium with 100 µg/mL NAT or 400 µg/mL HygB and knockouts were PCR verified at the 5′ flank with LCR031 or AJ60 and gene specific "5′ KO confirmation primer", at the 3′ flank with LCR323 or KA68 and gene specific "3′ KO confirmation primer" and for lack of the native locus with gene specific "orf confirmation" primer pairs (Supplementary Data 8).

For $P_{H3}$:*PDR9* strains, a NAT cassette-histone 3 promoter (CNAG_06745) fusion was synthesized commercially (Biomatik) and cloned into a pUC57 plasmid (NAT-$P_{H3}$, Supplementary Data 9). LCR443 and LCR444 primers were used to amplify a repair construct targeted to replace the native promoter of *PDR9* (CNAG_07799) with the NAT-$P_{H3}$ fusion. Two promoter-region specific sgRNAs were generated with primers LCR451-LCR454. Repair constructs were transformed with Cas9 DNA and two sgRNAs into H99. Transformants were selected on YPD medium with 100 µg/mL NAT and integration was PCR verified at the 5′ flank with LCR031 and LCR460 and at the 3′ flank with LCR323 and LCR459.

For *PDR9*-mNeonGreen fusion proteins, the mNeonGreen cassette was amplified from pBHM2406 (a gift from Hiten Madhani (Addgene plasmid # 173442; http://n2t.net/addgene:173442; RRID:Addgene_173442[54]) with primers LCR480 and LCR481 targeting the 3′ end of CNAG_07799 and removing the stop codon. The presence of C-terminal tag was confirmed with primers LCR455 and LCR323 and mNeonGreen expression verified by fluorescence microscopy.

### Competition assay
96-well plates were prepared with 200 µL YPD per well. Wells were inoculated with individual strains to be tested and the reference strain H99-mNeonGreen and grown statically overnight at 30 °C. Dilutions were performed to standardize input of unlabeled strains and H99-mNeonGreen in a 1:1 ratio, at $2 \times 10^4$ cells/mL final concentration. Co-cultures were grown at 30 °C for 24 h in ambient air or 5% $CO_2$ before analyzing on an Attune NxT Flow Cytometer with CytKick autosampler and Attune Cytometric software. Gating strategy (Supplementary Fig. 8) was optimized for single cells. mNeonGreen positive and negative populations were identified by histogram plot with 100,000 cells counted per sample. The percent of mNeonGreen negative cells in $CO_2$ conditions was divided by the percent of mNeonGreen negative cells in ambient conditions to determine the competitive fitness in $CO_2$ relative to ambient conditions. Statistical significance was determined by CHI square test in Microsoft Excel. If mutant strains were deficient in ambient conditions (below 10% of the population at 24 h) a two-sided Student's t-test was used to compare the population in $CO_2$ to ambient.

### Spot dilution assays
Cells from overnight cultures were washed twice with PBS prior to quantifying $OD_{600}$. Strains were diluted to an $OD_{600}$:1, followed by ten-fold serial dilutions. 3 µL from each dilution was spotted on agar plates and grown inverted at 30 °C or 37 °C in ambient air or at 5% $CO_2$ on RPMI 1640 with 165 mM MOPS, pH 7 plates alone or containing varying concentrations of rapamycin as indicated, with images captured at 48 h. For work with copper promoter-containing strains, 3 µL from each dilution was spotted on SC agar plates alone or containing copper

sulfate or bathocuproinedisulfonic acid (BCS) as indicated and grown inverted at 37 °C with images captured at 96 hours.

## RNA-Seq and differential expression analysis

Overnight cultures of indicated $CO_2$ sensitive or tolerant strains were washed and diluted to $7.5 \times 10^5$ cells/mL, 3 mL cultured per condition in a 6-well plate in RPMI 1640 medium with 165 mM MOPS, pH 7 for 24 hours at 37 °C in ambient air or 5% $CO_2$. Total RNA was isolated from harvested cells according to manufacturer instructions with an Invitrogen PureLink RNA mini-kit (catalog no. 12183018 A, ThermoFisher) with on-column DNAse treatment (catalog no. 12185010; Thermo-Fisher). Biological triplicates were harvested for each strain and condition. Total RNA (>2 μg per sample) was submitted to Azenta Life Sciences for standard RNA-Seq next-generation sequencing. The RNA samples were quantified using Qubit 2.0 Fluorometer (ThermoFisher) and RNA integrity was checked using TapeStation (Agilent Technologies). The RNA sequencing libraries were prepared using the NEB Next Ultra II RNA Library Prep Kit for Illumina using the manufacturer's instructions (New England Biolabs). Briefly, mRNAs were initially enriched with Oligod(T) beads. Enriched mRNAs were fragmented for 15 min at 94 °C. First-strand and second-strand cDNA were subsequently synthesized. cDNA fragments were end repaired and adenylated at 3' ends, and universal adapters were ligated to cDNA fragments, followed by index addition and library enrichment by PCR with limited cycles. The sequencing libraries were validated on the Agilent TapeStation (Agilent Technologies) and quantified by using Qubit 2.0 Fluorometer (ThermoFisher) as well as by quantitative PCR (KAPA Biosystems). The sequencing libraries were multiplexed and clustered onto a flowcell. After clustering, the flowcell was loaded onto the Illumina HiSeq instrument according to the manufacturer's instructions. The samples were sequenced using a 2 × 150 bp Paired End configuration. Image analysis and base calling were conducted by the HiSeq Control Software. Raw sequence data (.bcl files) generated from Illumina HiSeq was converted into fastq files and de-multiplexed using Illumina bcl2fastq 2.20 software. One mismatch was allowed for index sequence identification. Paired-end Illumina sequence read files were evaluated for quality and the absence of adaptor sequence using FastQC (https://www.bioinformatics.babraham.ac.uk/projects/fastqc/). Read files were mapped to *C. neoformans* reference genome H99 v48 (FungiDB) and gene transcript expression was quantified using HISAT2 and Stringtie[56]. Differential expression fold change, Wald test *p* values, and Benjamini-Hochberg adjustment for multiple comparisons were determined using DESeq2. Principle component analysis was performed on regularized log transformed gene counts to confirm the absence of batch effects[57].

## Protein extraction and Mpk1 western blot

Overnight cultures were diluted to an $OD_{600}$:0.1 in YPD, then grown to mid-log phase (4 h) at 37 °C, shaking at 200 rpm, with samples taken each hour. Protein was extracted in extraction buffer (10 mM HEPES [pH 7.4 to 7.9], 1.5 mM $MgCl_2$, 10 mM KCl, 1 mM dithiothreitol [DTT], 1× HALT protease and phosphatase inhibitor cocktail [catalog no. 1861280; ThermoFisher]) by five bead-beating cycles of 30 sec followed by 30 sec on ice per cycle. Debris and beads were pelleted before supernatant was recovered, and the protein concentration was quantified by Bradford assay. Protein (20 μg/lane) was loaded on a 10% SDS-PAGE gel and run at 80 V. Samples were transferred to nitrocellulose membrane for 1 h at 100 V, then the membrane was stained with Ponceau for 5 min at room temperature (RT), rinsed with distilled water and an image of loading acquired. The membrane was blocked with 5% bovine serum albumin (BSA) in tris-buffered saline with 0.1% tween 20 (TBST) for 1 h at RT, then incubated with 1:2,000 rabbit anti-p-p44 (phospho-p44/42 MAPK, catalog no. 4370; Cell Signaling) in 5% BSA/TBST overnight at 4 °C. The membrane was washed 3 times for 5 min with TBST, then incubated for 1 h at RT with 1:10,000 goat anti-rabbit horseradish peroxidase (HRP) (catalog no. STAR208P; Bio-Rad)

in 5% BSA/TBST. The membrane was washed 3 times for 5 min with TBST, then developed with chemiluminescent substrate (catalog no. 1705060; BioRad) and imaged on a myECL imager (ThermoFisher).

## Rim101-GFP pulldown and western blot

GFP-pulldown was performed as previously described[26]. Overnight cultures of H99 were washed twice in PBS and diluted to an $OD_{600}$:0.2 into RPMI 1640 + 165 mM MOPS, pH 7 and grown at 37 °C in ambient air or 5% $CO_2$ to an $OD_{600}$:1. 20 mL of culture was harvested and washed twice with water. Pellets were resuspended in 1 mL of ice-cold lysis buffer (50 mM Tris-HCl pH 7.4, 150 mM NaCl, 1% NP-40, 5 mM EDTA, 1 mM phenylmethanesulfonylfluoride (PMSF), 1× HALT protease and phosphatase inhibitor cocktail [catalog no. 1861280; ThermoFisher]). Cells were lysed by bead beating, with cell suspensions combined with 0.2 mL of 0.5 mm glass beads in a FastPrep-24 (MP Biomedicals) at top speed with 4 rounds of 45 s bead beating followed by 1 min on ice. Supernatants were transferred to new tubes and glass beads washed 2 times with 0.4 mL lysis buffer. Lysates were cleared by centrifugation at 18,000 x *g*, 4 °C, for 10 min. GFP-TRAP resin (catalog no. gta; Chromotek) was equilibrated in lysis buffer, then 10 μL was added to the cleared lysate and rocked at 4 °C overnight. GFP-TRAP resin was washed with 1 mL lysis buffer 3 times and protein was eluted in 30 μL 4X Laemmli sample buffer (catalog no. 1610747; Bio-Rad) with 2-mercaptoethanol by boiling for 5 min. The entire sample was loaded on a 10% SDS-PAGE gel and run at 80 V. Samples were transferred to a nitrocellulose membrane for 1 h at 100 V. The membrane was blocked with 5% BSA in tris-buffered saline with tween 20 (TBST) for 1 h at RT, then incubated with 1:1,000 mouse anti-GFP clones 7.1 and 13.1 (catalog no. 118144600001; Sigma) in 5% BSA/TBST overnight at 4 °C. The membrane was washed 3 times for 5 min with TBST, then incubated for 1 h at RT with 1:10,000 goat anti-mouse horseradish peroxidase (HRP) (catalog no. STAR207P; Bio-Rad) in 5% BSA/TBST. The membrane was washed 3 times for 5 min with TBST, then developed with chemiluminescent substrate (catalog no. 1705060; Bio-Rad) and imaged on a myECL imager (ThermoFisher).

## Nanostring and qRT-PCR

Overnight cultures of indicated strains were washed and diluted to 7.5 × $10^5$ cells/mL, 3 mL cultured per condition in a 6-well plate in RPMI 1640 medium with 165 mM MOPS, pH 7 for 24 h at 37 °C in ambient air or 5% $CO_2$. Total RNA was isolated from harvested cells according to manufacturer instructions with an Invitrogen PureLink RNA mini-kit with on-column DNAse treatment. Biological triplicates were harvested for each strain and condition. For Nanostring, 100 ng of RNA was hybridized to a custom Nanostring probe set (Supplementary Data 6) at 65 °C for 18 hours and quantified on a Nanostring Sprint nCounter. RCC files were imported into nSolver software to extract counts and evaluate quality control metrics. The mean of the negative control probes value plus 2 times the SD was subtracted from counts to provide a background threshold. Values below background were set to 1. After background subtraction, RNA counts were normalized by total count to the highest counts of any sample in comparison. Normalized counts were used to generate a heat map in Morpheus (https://software.broadinstitute.org/morpheus), hierarchically clustered by strain/condition and gene ID with one minus Pearson correlation, average linkage. For qRT-PCR, 1000 ng RNA was used for cDNA synthesis with iScript cDNA synthesis kit (catalog no. 1708891; Bio-Rad) according to manufacturer's instructions, then cDNA was diluted 1:5 with dd$H_2O$. qRT analysis was performed in 20 μL reactions using 2 μL of dilute cDNA per reaction with iQ SYBR Green Supermix (catalog no. 1708882; Bio-Rad). *pdr9* gene expression was measured with a Bio-Rad CFX Connect Real Time PCR Detection System (primers LCR461 and LCR462). Gene expression was normalized to actin expression (primers LCR225 and LCR226). Measurements were performed in technical duplicates with biological triplicates.

## Disk diffusion assays

Cells from overnight cultures were washed twice with PBS prior to quantifying concentration on a Countess II FL automated cell counter. Cells were diluted to $1 \times 10^7$ cells/mL to plate on RPMI. A sterile q-tip was saturated in the cell solution before streaking an entire plate to generate a lawn. When plates had dried, a sterile filter paper dot was placed in the center of the plate for a single treatment per plate or evenly spaced for multiple treatments. Treatments were added in 20 µL volumes at indicated final concentrations per disk: myriocin 8 µg/disk, fluconazole 40 µg/disk, aureobasidin 100 µg/disk, duramycin 200 µg/disk.

## Annexin V staining

Overnight cultures of *Cryptococcus* strains were washed in PBS and used to inoculate 25 mL cultures of RPMI 1640 medium with 165 mM MOPS, pH 7. Identical cultures for each strain were placed at 37 °C in ambient air or 5% $CO_2$ shaking at 200 rpm overnight. The following day 1–2 mL of the culture were collected and spun down to be washed with 1 mL of Annexin V binding buffer (10 mM HEPES-NaOH, pH 7.4, 140 mM NaCl, and 2.5 mM $CaCl_2$). The cell pellets were resuspended in 25 µl of Annexin V binding buffer and 1.5 µl of FITC-conjugated annexin V (catalog no. A13201, Life Technologies, Inc.) was added and incubated at RT for 15 min. The cells were pelleted and washed with 1 mL of Annexin V binding buffer. A final resuspension into 50 µl of Annexin V binding buffer was performed and cells were placed on a microscope slide with a coverslip. Confocal images (multiphoton laser scanning microscope (SP8; Leica Microsystem) LAS X software) of representative fields were taken for bright field and FITC fluorescence. All images were acquired using 488-nm excitation for the FITC signal with a 100X oil immersive objective lens. Image brightness and contrast were adjusted equivalently across samples in ImageJ for publication. Z-stacks were condensed into max projections and mean fluorescence intensity was calculated for a minimum of 100 cells for each condition using ImageJ.

## Fluorescence microscopy

Overnight cultures were diluted to an $OD_{600}$:0.1 and grown for 4 h to mid-log phase in YPD. Cells were washed, incubated with NucBlue Live ReadyProbes Reagent for 30 minutes according to manufacturer instructions (catalog no. R37605, ThermoFisher) and imaged on a confocal microscope (multiphoton laser scanning microscope (SP8; Leica Microsystem) LAS X software). All images were acquired using 488-nm excitation for mNeonGreen signal and the UV-laser for Hoescht signal with a 100X oil immersive objective lens. Image brightness and contrast were adjusted equivalently within each channel in ImageJ for publication.

## NBD-lipid uptake assays

Uptake of NBD labeled phosphatidylserine or phosphatidylethanolamine was examined based on methods described in Viet et al.[58]. Briefly, overnight cultures were diluted to an $OD_{600}$:0.2 in RPMI 1640 medium with 165 mM MOPS, pH 7 and grown for 18 hours at 30 °C in ambient or 5% $CO_2$ shaking conditions. Cells were harvested, washed and resuspended to $OD_{600}$:8 in PBS. 250 µL of cells were aliquoted into a round-bottom 96-well plate for each condition to be tested in triplicate, then incubated at 30 °C for 10 minutes. 1.5 µL of 10 mM NBD-PS (catalog no. 810192P, Avanti Polar Lipids) or 10 mM NBD-PE (catalog no. 810151P, Avanti Polar Lipids) was added to cells and incubated for 30 minutes at 30 °C. After incubation, cells were washed three times with PBS + 4% BSA to extract free NBD-lipids and examined by flow cytometry on an Attune NxT Flow Cytometer with CytKick autosampler and Attune Cytometric software. Gating strategy was optimized for single cells. 100,000 single cells were acquired per sample. Mean fluorescence intensity was determined for each sample.

## Melaninization assay

Overnight cultures of H99 or $P_{H3}$:*PDR9* were washed twice in sterile PBS before resuspending in PBS and quantifying $OD_{600}$. Strains were diluted to an $OD_{600}$:1, followed by ten-fold serial dilutions. Three µL from each dilution was spotted onto solid L-DOPA media (7.6 mM L-asparagine monohydrate, 5.6 mM glucose, 22 mM $KH_2PO_4$, 1 mM $MgSO_47H_2O$, 0.5 mM L-DOPA, 0.3 mM thiamine-HCl, 20 nM biotin, 2% agar), grown inverted at 30 °C and imaged at indicated time points.

## Capsule production

Overnight cultures of H99 or $P_{H3}$:*PDR9* were washed twice in sterile PBS before resuspending in PBS and quantifying concentration on a Countess II FL automated cell counter. Cells were diluted to $1.25 \times 10^5$ cells/mL in RPMI 1640 with 165 mM MOPS, pH 7 and grown for 24 h at 37 °C in ambient air or 5% $CO_2$, then prepared for microscopy by concentrating cells and counterstaining with India ink. Images were captured using a Rebel microscope at 60X magnification. At least 50 cells were quantified per condition and processed in ImageJ software to measure capsule.

## Animal experiments

Six- to eight-week-old female CD1 mice were obtained from Charles River Laboratory. Animals were housed with 5 mice per cage, under a 12 hr dark/12 hr light cycle at ambient temperature (20–24 °C), 30–70% humidity. Prior to infection, overnight cultures of *C. neoformans* strains were washed with sterile saline three times and resuspended in saline. For the coinfection experiment, five CD1 mice per group were infected intranasally with a 1:1 ratio of WT and mutant cells ($5 \times 10^4$ cells per mouse). The ratio of WT and mutant cells was confirmed by plating serial dilutions of the inoculums on YNB and YNB with 100 µg/mL of NAT to measure CFU. At DPI 14, the mice were sacrificed, and the lung, brain, and left kidney, were dissected. For fungal burden quantifications, dissected organs were homogenized in 2 mL of cold sterile PBS using an IKA-T18 homogenizer. Homogenized organs were serially diluted, plated onto YNB and YNB with 100 µg/mL of NAT, and incubated at 30 °C for two days before counting CFUs.

For the survival study, ten CD1 mice per group were infected intranasally with $1 \times 10^4$ fungal cells. All mice were used for the survival study, and fungal burden was examined for five mice at the time of termination or at DPI 86. Fungal burden quantification was performed as described above.

## Phagocytosis assay

J774A.1 murine macrophage cells (catalog no. CB_91051511, ATCC) were seeded into a T25 flask at 40% confluence and incubated at 37 °C with 5% $CO_2$ for 48 h until 80 to 90% confluent, approximately $7.8 \times 10^6$ cells/flask. Overnight cultures of H99 or $P_{H3}$:*PDR9* were washed twice with PBS, before quantifying concentration on a Countess II FL automated cell counter. Cells were diluted to $3.12 \times 10^8$ cells/mL in PBS. Each strain was opsonized with mAb 18b7 (a generous gift from A. Casadevall[59]) at 2 µg/mL final concentration at 37 °C for 1 hour with rotation. After opsonization, 250 µL of cells were added to each flask of J774 cells for an MOI of 10. Triplicate flasks were prepared for each strain, then incubated for 2 h and 40 min and washed to remove non-phagocytosed cells. Macrophage cells were lysed with ice-cold 0.1% triton X in PBS for 10 min while rocking. For plating, cells were serially diluted 10-fold down to 1:1000. Four dilutions (full down to 1:1000) were plated on YPD with $3 \times 10$ µL spots per dilution and grown for 48 h at 30 °C before counting CFU.

## Statistical analysis

Data were graphed and analyzed in GraphPad Prism 9 software or Microsoft Excel. Statistical analysis and graph descriptions, including *P* values are provided in figure legends.

**Reporting summary**

Further information on research design is available in the Nature Portfolio Reporting Summary linked to this article.

## Data availability

The RNA-Seq data discussed in this publication have been deposited in NCBI's Gene Expression Omnibus[60] and are accessible through GEO Series accession number GSE 241788. RNA-Seq data was analyzed with *C. neoformans* H99 genome v48 available on FungiDB. The data supporting the findings of this work are provided within the article and supplementary files. Source data are provided with this paper. Materials generated in this study will be made available upon request. Source data are provided with this paper.

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

## Acknowledgements
This work was supported by the following grants from NIAID: 5R01AI147541 (D.J.K. and X.L.) and T32AI007511 (A.J.J.). The authors thank Andy Alspaugh and Connie Nichols (Duke) for sharing protocols and strains for experiments with Rim101. We thank Chaoyang Xue (Rutgers) for providing the P$_{CTR4}$-*CHO1* strain and Jim Kronstad (British Columbia) for providing the *cdc50Δ* strains. We also thank Scott Moye-Rowley (Iowa) for helpful discussions regarding ABC transporter biology and Rohan Wakade for assistance with fluorescence microscopy.

## Author contributions
L.C.R.: experimentation, data analysis, figure construction, writing of draft and editing manuscript. A.J.J.: experimentation, data analysis, figure construction. B.J.C.: experimentation, data analysis. M.A.S.: supervision, data analysis. X.L.: conception, data analysis, supervision, funding, editing of manuscript. D.J.K.: conception, data analysis, supervision, funding, writing and editing of manuscript.

## Competing interests
The authors declare no competing interest.
