## [Peer Review File · Nature Communications]

Cryptococcus neoformans adapts to the host environment through TOR-mediated remodeling of phospholipid asymmetryREVIEWER COMMENTS

Reviewer #1 (Remarks to the Author):

This paper describes the involvement of the TOR signaling pathway in *C. neoformans*' ability to adapt to host CO₂ levels. By using a combination of genetic screening and transcriptional profiling, the authors have identified specific components of the TOR signaling pathway that are essential for the fungus' tolerance to CO₂. In addition, transcriptional profiling revealed a set of highly expressed genes in CO₂-sensitive environmental strains, including an ABC transporter. The authors further demonstrate that the overexpression of this ABC transporter increases susceptibility to the PE-binding toxin, duramycin. Additionally, exposure of *C. neoformans* cells to CO₂ led to higher Annexin V binding, which was reduced upon overexpression of the ABC transporter. Based on these findings the authors conclude that plasma membrane PS asymmetry is regulated in the responds to CO₂. Although the article is of general interest and the authors show for the first time that the TOR pathway is involved in the cellular adaptation to CO₂ levels, the manuscript does not appear to be complete and requires further clarification and additional experiments.

Key issues with the manuscript are delineated below:

(1) The authors' findings regarding changes in lipid asymmetry in response to overexpression and/or increased CO₂ levels are not supported by strong evidence. Alternative explanations for the observed results have not been adequately excluded. For instance, a higher aminophospholipid content of the plasma membrane could lead to an overall increase in the levels of exposed PS and/or PE, without actual changes in lipid asymmetry. Additionally, variations in the sphingolipid and sterol content of the plasma membrane could impact the binding of both probes. Changes in the thickness of the capsule (fig. S9c) could also affect the binding of the probes. Therefore, the current experiments do not allow for a conclusive evaluation of changes in lipid asymmetry and the involvement of a putative floppase. No attempts have been made by the authors to measure floppase activities by assays based on fluorescent lipids.

(2) The results obtained from the overexpression of the ABC transporter require essential controls to confirm if the effect is based on a catalytically active transporter. Additionally, it is crucial to verify whether the transporter is localized at the plasma membrane. The mechanism of action of the transporter as a floppase also needs to be elucidated, as it is unclear how it can increase the levels of outer leaflet PE while simultaneously elevating inner leaflet PS levels. Further investigations with appropriate controls and techniques are necessary to address these questions.

Additional issue to be addressed to enhance the clarity and rigor of the study:

- The rationale for performing some experiments at 37 and some at 30 degrees and using 5% or 10% CO₂ should be clarified.
- Enriched nuclear localization should be confirmed by nuclear staining using, for instance, DAPI (Fig. S3).
- Dead cells should be excluded by staining with, for example, propidium iodide (Fig. 6F).
- The plate assay results (Fig. S2) appear to contradict the fitness assay results for some deletion mutants. Please clarify the reasons behind this discrepancy in the text.
- The elevated TOR pathway should show increased rapamycin sensitivity, which needs to be clarified in paragraph lines 159-162.
- The mpk1 blot should be repeated with mpk1 AB for the total amount to quantify the phosphorylated/unphosphorylated ratio (Fig. 4A).
-
- The size of unprocessed and processed rim101-gfp should be indicated in Fig. 4D.
- ARV1 is mentioned in the text (line 257) but not in the figure, which needs to be addressed.
- Lat1 is not a suitable name since it is already used for human L-type amino acid transporter 1. A new name should be proposed.
- In Fig. 5B, it should be discussed that the H99 strain shows the highest percentage increase in lat1 expression.
- Line 247 should briefly explain the histone 3 promoter system.
- In Fig. S6, Lat1 is even more highly expressed in CO₂ conditions, even though it is already normalized to the corresponding H99 value, which needs to be discussed.

- It should be clarified that myriocin affects sphingolipid, not phospholipid synthesis (line 284).
- The mechanism proposed for the effect of plasma membrane asymmetry on CO₂ tolerance should be explained (line 294).
- It should be addressed why H99 does not show higher sensitivity when shifted to higher CO₂, even though Lat1 is highly expressed in it (Fig. 6A).
- The study does not provide sufficient evidence to claim that Lat1 is a PE floppase (line 301).
- The hypothesis proposed in lines 435/436 needs to be elaborated on to enhance its clarity.
- It should be clarified that Cdc50 is not a flippase but the beta subunit (line 440).
- The mechanism of how a floppase reduces PS exposure (lines 448-453) needs to be explained.
- Other possible direct and indirect mechanisms should be discussed (line 461).
- Statistical data analysis, including error bars, should be included in Fig. 4C, 5C, 6B, and 6D to enhance the rigor of the study.

Reviewer #2 (Remarks to the Author):

In this manuscript, the authors build upon their prior important observations regarding an association between CO₂-tolerance and virulence in human fungal pathogenic *Cryptococcus* isolates. These are compelling studies since mammalian pathogens must be able to survive elevated CO₂ concentrations to effectively parasitize the host. They cite prior data that CO₂ tolerance in *Cryptococcus* is independent of responses to elevated pH. They find that some defined stress response pathways promote CO₂ tolerance, while other stress response pathways must be suppressed for maximal survival in high CO₂. They interpret these composite data that infectious microorganisms like *C. neoformans* must maintain a balance between activation and suppression of different stress response pathways in the setting of infection.

In the current manuscript, they screen two collections of strains with loss-of-function mutations in protein kinases and transcription factors to assess for altered CO₂ tolerance. By characterizing predicted functions of genes mutated in each mutant strain, the authors identify three main pathways likely to be involved in CO₂ sensing and response. One is the TOR pathway (further supported by follow-up studies using the TOR kinase inhibitor rapamycin). They also identify a pathway including the Gat201/Gat204 axis and the Yap1 protein. The identification of multiple genes in this single pathway (concordant phenotypes in corresponding mutants) further supports the association between Yap1 signaling and CO₂ tolerance. They also identify a fungal-specific stress response pathway (Rim) previously found to control growth at elevated pH.

It is interesting that some of their major findings are very similar to those in model yeasts such as *S. cerevisiae* (e.g., TOR's role in phospholipid asymmetry), while other signaling paradigms are distinct from work in model systems (signals leading to TOR activation).

The authors also use comparative transcriptional profiling to identify core genes required for TOR pathway inhibition (using rapamycin) or Rim pathway inactivation. They find that the TOR pathway is activated by elevated CO₂, and that the CWI pathway (characterized by Mpk1 phosphorylation) is "blunted" in activation by CO₂. Also, their data suggest that reduced Rim pathway activity helps growth in elevated CO₂ (opposite to growth at elevated pH). Therefore, reduced activity of two stress-response pathways (CWI and Rim) seems to paradoxically favor growth in CO₂. They link their complex data sets by suggesting that inactivation of Rim signaling activates TOR signaling, which in turn favors CO₂ tolerance (a visual model might be helpful given multiple interacting pathways)

The investigators also characterize a protein that they suggest is a floppase involved in membrane asymmetry. These results are based on previous studies in *S. cerevisiae* demonstrating that membrane asymmetry is in part mediated by TOR signaling. Although very interesting, the LAT1 experiments are perhaps the least definitive in terms of concrete interpretations. For example, there are many reasons that a single genetically altered strain (with constitutive LAT1 expression) might have decreased virulence: the experiments would be strengthened if two independent LAT1 over-expressing strains were tested, at least for some of the studies in altered virulence. Also comparing lung fungal burdens at two different time points after infection is difficult to interpret. The data presentation in Fig 7 suggesting a direct comparison (as shown in the graphs) is

confusing. The basic experimental plan for the virulence experiments is otherwise very good.

Minor points:

- 1) The western blot of Rim101 cleavage and processing demonstrates decreased intensity of many of the larger MW bands in the sample incubated in higher CO₂. The authors conclude that Rim101 is degraded in this condition. Are the cells equally lysed in these two conditions, or could CO₂-induced cell changes (cell wall, capsule, etc) alter total protein recovery? Any controls for total protein?
- 2) The sentence beginning in line 345 is quite speculative, and uncharacteristic of the careful interpretation of the rest of the paper.

Reviewer #3 (Remarks to the Author):

In this paper, the authors screened *Cryptococcus neoformans* kinase and transcription factor mutant libraries for their sensitivity to host concentration in CO₂. They thus identified 21 kinase mutants and several TF regulating *C. neoformans* growth under this condition. The authors identified the TOR pathway as playing a major positive regulator role whereas the the RIM and CWI pathways have opposite effects. To support this hypothesis, they studied both CO₂ resistant and sensitive environmental isolates for their resistance to CO₂/rapamycin, they performed RNA-Seq experiments of several mutant strains, in the presence of CO₂ and/or Rapamycin as well as several western blot experiments. They also used Nanostring experiments to identify the keys targets of this regulatory pathway, which conducted them to study a protein encoded by the gene CNAG_07799 which seems to be a floppase regulating virulence in this pathogenic yeast. Overall, the paper is very well written, and the experiments well presented. Most of the conclusions are well supported by the results of this experiments although few questions remain.

- 1) Although I am convinced that the TOR pathway positively regulates CO₂ resistance whereas RIM and MPK are doing the opposite, it is still difficult for the reader to have a synthetic view of this regulation including the action of the TF identified. Maybe a schematic model would help.
- 2) Figure 1. Swe102 mutant should be deleted from figure 1D as the reconstruction of the mutation does not restore the observed phenotype
- 3) Figure 2. The effect of rapamycin on WT under CO₂ is not clear by dot assays, may be the authors should use the type of assay they used for the screening to make their point. The same for the environmental isolates
- 4) RNA-Seq. The authors reported little effect of CO₂ after after 4 and 8 hours (lines 171-172). Indeed, the presence of CO₂ 5% should be transduced immediately. Why waiting 24h? Does the short-term effect depend on tor? It is transient? Can the effect be seen at 24h depend on something else? For instance, a compensatory mechanism. It might be important here to describe and to analyse the genes regulated after shorter time.
- 5) Figure 3D: the position of the labels is misleading. The percentage analysis in the text are interesting but not the one in the figure. These last ones should be deleted.
- 6) Figure 4. Phosphorylation of MPK1: it would be easier to read if quantification of the band intensity was done. Furthermore, in the RIM101 blot, the size of the processed and unprocessed protein should be indicated by an arrow. A control using an unrelated protein should be used to make sure that this result is not due to a general protein degradation. The scheme for RIM101 might be good but should be improved (for instance, the size and the color of the peptide released from RIM101 upon proteolyze is not the same as in the preprocessed protein.
- 7) The selection of the 118 genes used for the nanostring experiment is not justified. Are they the most altered, the most expressed, other criteria?
- 8) The best homolog of CNAG_07799 is not PDR5 but SNQ2 although it is not reciprocal. PDR5 is only the third best homolog. The homology is high not modest as written line 292. Although I agree that CNAG_07799 seems to ask as a Floppase and cannot be named PDR5 or SNQ2, it cannot be named neither LAT1 which is *S. cerevisiae* designs a Dihydrolipoamide acetyltransferase (source SGD). This part should be changed.
- 9) Figure 6: Disk assays are good but these experiments seem to have been performed only once and not statistical analysis of the difference of the zone of clearance size is given. This should be corrected.

Ristow et al – Reviewer response

We thank the reviewers for their constructive comments. Although the reviewers raised some points regarding the screening, transcriptional, and Mpk1/Rim101 focused experiments, the most extensive comments were in regard to our characterization of the ABC/PDR transporter and its molecular function. As detailed here and in the manuscript, we have performed a number of additional experiments to further characterize its function and effects on membrane homeostasis. In addition, we have adopted the previously published name for this ABC transporter and refer to it as *PDR9*. Although these experiments support our initial assertion that *PDR9* may be a phospholipid floppase, the demonstration of this activity is challenging and, indeed, essentially no ABC transporter has been conclusively demonstrated to function as a phospholipid floppase. Therefore, we were naïve to put forth such a strong assertion that *PDR9* is a floppase. Accordingly, we both provide additional experimental data to characterize the effect of *PDR9* on membrane homeostasis and lipid asymmetry and a more nuanced interpretation of those results. Additionally, we have performed additional experiments and controls with regard to our finding that CO₂ increases PM outer leaflet PS and we feel this conclusion can be made with even more confidence.

We have also responded to other comments and queries as detailed below.

Overall, we feel the manuscript is much stronger and thank the reviewers for their contribution to its improvement.

Reviewer #1 (Remarks to the Author):

This paper describes the involvement of the TOR signaling pathway in *C. neoformans*' ability to adapt to host CO₂ levels. By using a combination of genetic screening and transcriptional profiling, the authors have identified specific components of the TOR signaling pathway that are essential for the fungus' tolerance to CO₂. In addition, transcriptional profiling revealed a set of highly expressed genes in CO₂-sensitive environmental strains, including an ABC transporter. The authors further demonstrate that the overexpression of this ABC transporter increases susceptibility to the PE-binding toxin, duramycin. Additionally, exposure of *C. neoformans* cells to CO₂ led to higher Annexin V binding, which was reduced upon overexpression of the ABC transporter. Based on these findings the authors conclude that plasma membrane PS asymmetry is regulated in the responds to CO₂. Although the article is of general interest and the authors show for the first time that the TOR pathway is involved in the cellular adaptation to CO₂ levels, the manuscript does not appear to be complete and requires further clarification and additional experiments.

Key issues with the manuscript are delineated below:

(1) The authors' findings regarding changes in lipid asymmetry in response to overexpression and/or increased CO₂ levels are not supported by strong evidence. Alternative explanations for the observed results have not been adequately excluded. For instance, a higher aminophospholipid content of the plasma membrane could lead to an overall increase in the levels of exposed PS and/or PE, without actual changes in lipid asymmetry. Additionally, variations in the sphingolipid and sterol content of the plasma membrane could impact the binding of both probes. Changes in the thickness of the capsule (fig. S9c) could also affect the binding of the probes. Therefore, the current experiments do not allow for a conclusive evaluation of changes in lipid asymmetry and the involvement of a putative floppase. No

attempts have been made by the authors to measure floppase activities by assays based on fluorescent lipids.

Response: We have performed three experiments to address the possibility that higher phospholipid content of the plasma membrane could lead to increased outer membrane PS/PE without changing asymmetry. First, we performed PS uptake assays using fluorescently labelled PS (as the review suggested). As shown in Fig. 6E, there is a clear and statistically significant reduction in PS uptake in CO₂ exposed cells relative to those at ambient. This is consistent with a change in asymmetry. Second, we used a strain in which the PS synthase (*CHO1*) is under the control of a copper regulated promoter to suppress its expression. If the cells were increasing total PS to compensate for CO₂, then reducing expression of *CHO1* would reduce fitness; as shown in Fig. S5B, the strain shows reduced growth in ambient conditions, indicating that *CHO1* is repressed, but there is no significant reduction in growth in CO₂. Thirdly, *Cdc50* has been shown by the Xue lab to be required for PS asymmetry through its regulation of P4-ATP flippases and the deletion mutant shows increased outer leaflet PS staining via Annexin V staining. The *cdc50Δ* mutant shows significant resistance to CO₂ (Fig. 6F), consistent with the conclusion that increased outer leaflet PS is part of the adaptive response to CO₂.

With respect to variation in lipid or sterol content confounding the Annexin V assay, this is possible in principle. The assay has been used by many investigators in many systems including multiple uses in fungi and yeast. We can find no indications that others, including the Xue and Kronstad labs (ref. 34&35), have observed such effects or designed approaches to control for that potential confounding effect. In terms of the capsule interfering with Annexin V binding, we can observe signal in WT cells in ambient conditions and a significant increase in Annexin V staining in capsule inducing conditions (+CO₂, Fig.6c&d). Therefore, we feel this assay is an appropriate approach to the assessment of outer leaflet PS.

We attempted cellular floppase assays based on PE and PS using methods used in *S. cerevisiae*. Specifically, cells were loaded with labeled PS/PE at low temperature and then shifted to 30°C and intracellular fluorescence over time followed by flow cytometry. However, we were not able to confidently observe export of the lipids due to low signal. As noted above, we have substantially changed our assertions regarding the molecular function of *PDR9*.

(2) The results obtained from the overexpression of the ABC transporter require essential controls to confirm if the effect is based on a catalytically active transporter. Additionally, it is crucial to verify whether the transporter is localized at the plasma membrane. The mechanism of action of the transporter as a floppase also needs to be elucidated, as it is unclear how it can increase the levels of outer leaflet PE while simultaneously elevating inner leaflet PS levels. Further investigations with appropriate controls and techniques are necessary to address these questions.

Response: We have generated a *PDR9*-GFP allele under the control of both the endogenous promoter and the P_{H3} promoter. The fusion protein is functional based on the ability of the P_{H3}-*PDR9*-GFP strain to sensitize cells to CO₂ (Fig. S4A). As shown in Fig. S4B, *PDR9* does not show the canonical localization of PDR transporters to the plasma membrane. Instead, it is localized to intracellular puncta that are most consistent with the late Golgi/early endosome based on *S. cerevisiae*. As we note in the text (line 549-555), localization to this compartment has been observed for mammalian ABC transporters that have broad effects on membrane homeostasis (ref. 48). Clearly, a significant amount of biochemistry, cell biology, and biophysics will be required to define the molecular mechanisms of Pdr9 function and this is beyond the scope of the current report. We feel that mutagenesis falls within those studies. We have expanded our discussion of the different possibilities for how *PDR9* affects membrane biology

and CO₂ sensitivity and no longer claim that it is a bona fide floppase. It clearly has a complex effect on membrane biology.

Additional issue to be addressed to enhance the clarity and rigor of the study:

1. The rationale for performing some experiments at 37 and some at 30 degrees and using 5% or 10% CO₂ should be clarified.

Response: We performed the screen at 30°C in order to avoid mutants with temperature sensitive phenotypes confounding the screen. We have explained this in line 106. We no longer report data for 5 and 10% CO₂; that figure was used in error.

2. Enriched nuclear localization should be confirmed by nuclear staining using, for instance, DAPI (Fig. S3).

Response: We have removed that figure and the new microscopy with *PDR9*-GFP is done with DAPI to show nucleus.

3. Dead cells should be excluded by staining with, for example, propidium iodide (Fig. 6F).

Response: CO₂ is fungistatic toward H99 as we have previously reported (ref. 10) and the hypersensitive mutants have reduced PS staining; dead cells show increased PS staining so we cannot explain the differences in PS staining observed in that way.

4. The plate assay results (Fig. S2) appear to contradict the fitness assay results for some deletion mutants. Please clarify the reasons behind this discrepancy in the text.

Response: Thank you for pointing that out. Two mutants (*sch9Δ* and *bwc2Δ*) have phenotypes in the fitness assay but not on plates. The competition assay is likely to be more sensitive since it is a measure of relative growth rate while the spot dilution assay is driven more by final cell density. We have added a note about this to line 122-123.

5. The elevated TOR pathway should show increased rapamycin sensitivity, which needs to be clarified in paragraph lines 159-162.

Response: Thank you for pointing this out. We have reworded this section. The TF mutants would be resistant to rapamycin if they have elevated TOR activity as a compensatory response to the alterations in gene expression. This is confirmed in the case of the *rim101Δ* mutant as described in that section.

6. The mpk1 blot should be repeated with mpk1 AB for the total amount to quantify the phosphorylated/unphosphorylated ratio (Fig. 4A).

Response: Ideally, this would be possible. The original clones of the monoclonal antibodies for human ERK cross reacted with *C. neoformans* (and other yeast) Mpk1. However, that is no longer the case. The Bahn lab attempted to generate an antibody but was unsuccessful. Therefore, a review of the literature of studies on *C. neoformans* indicates that investigators have done as we have.

7. The size of unprocessed and processed rim101-gfp should be indicated in Fig. 4D.

Response: We agree and have done so.

8. ARV1 is mentioned in the text (line 257) but not in the figure, which needs to be addressed.

Response: We now refer to this gene in both text and figure.

9. Lat1 is not a suitable name since it is already used for human L-type amino acid transporter

1. A new name should be proposed.

Response: We agree and have revised; see introduction.

10. In Fig. 5B, it should be discussed that the H99 strain shows the highest percentage increase in *lat1* expression.

Response: Thank you for bringing up this point. Our data are more consistent with the absolute expression being important for CO₂ tolerance and not fold change in expression relative to ambient conditions. We now make that distinction clear at line 274-278.

11. Line 247 should briefly explain the histone 3 promoter system.

Response: We now indicate that it has been used previously to overexpress genes and a reference is given.

12. In Fig. S6, *Lat1* is even more highly expressed in CO₂ conditions, even though it is already normalized to the corresponding H99 value, which needs to be discussed.

Response: We note that in the text but indicate that it is less than 1.5X. Line 286.

13. It should be clarified that myriocin affects sphingolipid, not phospholipid synthesis (line 284).

Response: We have corrected that; thank you for noticing the error.

14. The mechanism proposed for the effect of plasma membrane asymmetry on CO₂ tolerance should be explained (line 294).

Response: At this point in the manuscript, we based this hypothesis on the fact that the TOR pathway regulates PM asymmetry in *S. cerevisiae* and CO₂ tolerance in *C. neoformans*. The section referred to is the premise for testing CO₂ and the mutants for alterations in asymmetry. Line 308-314.

15. It should be addressed why H99 does not show higher sensitivity when shifted to higher CO₂, even though *Lat1* is highly expressed in it (Fig. 6A).

Response: This is related to point 10. Although *PDR9* is induced in H99, the absolute levels are below those of sensitive strains and the *PDR9* mutant. H99 has a mild growth defect in CO₂ relative to ambient and this could contribute to that. See line 274-278.

16. The study does not provide sufficient evidence to claim that *Lat1* is a PE floppase (line 301).

Response: As discussed above, we have revised that claim.

17. The hypothesis proposed in lines 435/436 needs to be elaborated on to enhance its clarity.

Response: Thank you for noting this. We have revised it to read: In general, PS supports membrane fluidity whereas increased CO₂ concentrations are reported to decrease the fluidity of microbial membranes (44). Based on these opposing biophysical effects of PS and CO₂ on biological membranes, one potential explanation for the increased CO₂ fitness of *C. neoformans* cells is that the increased outer leaflet PS promotes membrane fluidity to counter the membrane rigidifying effects of elevated CO₂. Line 506-513.

18. It should be clarified that *Cdc50* is not a flippase but the beta subunit (line 440).

Response: We have corrected that error, see line 516-518.

19. The mechanism of how a floppase reduces PS exposure (lines 448-453) needs to be explained.

Response: We explain in multiple sections of the manuscript that the effect of *PDR9* is most likely indirect based on current knowledge of the regulators of PM asymmetry.

20. Other possible direct and indirect mechanisms should be discussed (line 461).

Response: We have added additional discussion. Please see lines 514-548.

21. Statistical data analysis, including error bars, should be included in Fig. 4C, 5C, 6B, and 6D to enhance the rigor of the study.

Response: We have revisited all figures to make sure those details are complete.

Reviewer #2 (Remarks to the Author):

In this manuscript, the authors build upon their prior important observations regarding an association between CO₂-tolerance and virulence in human fungal pathogenic *Cryptococcus* isolates. These are compelling studies since mammalian pathogens must be able to survive elevated CO₂ concentrations to effectively parasitize the host. They cite prior data that CO₂ tolerance in *Cryptococcus* is independent of responses to elevated pH. They find that some defined stress response pathways promote CO₂ tolerance, while other stress response pathways must be suppressed for maximal survival in high CO₂. They interpret these composite data that infectious microorganisms like *C. neoformans* must maintain a balance between activation and suppression of different stress response pathways in the setting of infection.

In the current manuscript, they screen two collections of strains with loss-of-function mutations in protein kinases and transcription factors to assess for altered CO₂ tolerance. By characterizing predicted functions of genes mutated in each mutant strain, the authors identify three main pathways likely to be involved in CO₂ sensing and response. One is the TOR pathway (further supported by follow-up studies using the TOR kinase inhibitor rapamycin). They also identify a pathway including the Gat201/Gat204 axis and the Yap1 protein. The identification of multiple genes in this single pathway (concordant phenotypes in corresponding mutants) further supports the association between Yap1 signaling and CO₂ tolerance. They also identify a fungal-specific stress response pathway (Rim) previously found to control growth at elevated pH.

It is interesting that some of their major findings are very similar to those in model yeasts such as *S. cerevisiae* (e.g., TOR's role in phospholipid asymmetry), while other signaling paradigms are distinct from work in model systems (signals leading to TOR activation).

The also use comparative transcriptional profiling to identify core genes required for TOR pathway inhibition (using rapamycin) or Rim pathway inactivation. They find that the TOR pathway is activated by elevated CO₂, and that the CWI pathway (characterized by Mpk1 phosphorylation) is "blunted" in activation by CO₂. Also, their data suggest that reduced Rim pathway activity helps growth in elevated CO₂ (opposite to growth at elevated pH). Therefore, reduced activity of two stress-response pathways (CWI and Rim) seems to paradoxically favor growth in CO₂. They link their complex data sets by suggesting that inactivation of Rim signaling activates TOR signaling, which in turn favors CO₂ tolerance (a visual model might be helpful given multiple interacting pathways).

Response: Thank you for the suggestions. We tried a variety of schemes but they seemed to make things more confusing. We have added a discussion of the link between Rim101 and Cdc50 described by Brown et al. (ref. 45).

The investigators also characterize a protein that they suggest is a floppase involved in membrane asymmetry. These results are based on previous studies in *S. cerevisiae* demonstrating that membrane asymmetry is in part mediated by TOR signaling. Although very interesting, the LAT1 experiments are perhaps the least definitive in terms of concrete interpretations. For example, there are many reasons that a single genetically altered strain

(with constitutive LAT1 expression) might have decreased virulence: the experiments would be strengthened if two independent LAT1 over-expressing strains were tested, at least for some of the studies in altered virulence.

Response: We have included data from an PH3-PDR9 strain generated in a second strain background and it shows the same phenotype. Fig. S3D and lines 290-291. Given the consistent in vitro phenotypes, we did not repeat animal experiments in the interest of limiting animal use. We also examined the interactions of the PH3-PDR9 strain with macrophages as a potential additional phenotype related to virulence. As shown in Fig. 7F, these strains have reduced macrophage phagocytosis which is consistent with their increased capsule but does not explain their reduced virulence. We feel that is still best explained by reduced CO₂ tolerance. Line 415-421.

Also comparing lung fungal burdens at two different time points after infection is difficult to interpret. The data presentation in Fig 7 suggesting a direct comparison (as shown in the graphs) is confusing. The basic experimental plan for the virulence experiments is otherwise very good.

Response: Our intent was to compare fungal burden in the same at time of sacrifice for H99 and directly compare to surviving *P_{H3}-PDR9*. If we group by strain, then the important comparisons are between two graphs instead of within one graph. So, we prefer to keep the arrangement as initially presented. However, we are open to other suggestions.

Minor points:

1) The western blot of Rim101 cleavage and processing demonstrates decreased intensity of many of the larger MW bands in the sample incubated in higher CO₂. The authors conclude that Rim101 is degraded in this condition. Are the cells equally lysed in these two conditions, or could CO₂-induced cell changes (cell wall, capsule, etc) alter total protein recovery? Any controls for total protein?

Response: As shown in the total protein blot for the Mpk1 western, there is no difference in total protein amounts in extracts of cells in ambient and 5% CO₂. We added equal amounts of total protein to the IP for the Rim101 and that is now indicated in the legend.

2) The sentence beginning in line 345 is quite speculative, and uncharacteristic of the careful interpretation of the rest of the paper.

Response: It has been removed.

Reviewer #3 (Remarks to the Author):

In this paper, the authors screened *Cryptococcus neoformans* kinase and transcription factor mutant libraries for their sensitivity to host concentration in CO₂. They thus identified 21 kinase mutants and several TF regulating *C. neoformans* growth under this condition. The authors identified the TOR pathway as playing a major positive regulator role whereas the the RIM and CWI pathways have opposite effects. To support this hypothesis, they studied both CO₂ resistant and sensitive environmental isolates for their resistance to CO₂/rapamycin, they performed RNA-Seq experiments of several mutant strains, in the presence of CO₂ and/or Rapamycin as well as several western blot experiments. They also used Nanostring experiments to identify the keys targets of this regulatory pathway, which conducted them to study a protein encoded by the gene CNAG_07799 which seems to be a floppase regulating virulence in this pathogenic yeast. Overall, the paper is very well written, and the experiments well presented. Most of the conclusions are well supported by the results of this experiments although few questions remain.

1) Although I am convinced that the TOR pathway positively regulates CO₂ resistance whereas RIM and MPK are doing the opposite, it is still difficult for the reader to have a synthetic view of this regulation including the action of the TF identified. Maybe a schematic model would help.
Response: Please see comment of review 2. Our attempts at a synthesis scheme all seemed overly complex and confusing.

2) Figure 1. Swe102 mutant should be deleted from figure 1D as the reconstruction of the mutation does not restore the observed phenotype
Response: This point is well taken. However, we respectfully disagree. We feel it is important to show negative data to provide the reader with an idea of the likely false positive rate of the screen since we did not retest all mutants.

3) Figure 2. The effect of rapamycin on WT under CO₂ is not clear by dot assays, may be the authors should use the type of assay they used for the screening to make their point. The same for the environmental isolates
Response: Indeed, some of the phenotypes are subtle. Unfortunately, we cannot compare strains to themselves plus and minus drug/CO₂ using competitive fitness. The most clear-cut phenotypes are the increased rapamycin activity in CO₂ and the effect of rim101 which we follow up. We can remove the other mutants if this seems more appropriate.

4) RNA-Seq. The authors reported little effect of CO₂ after after 4 and 8 hours (lines 171-172). Indeed, the presence of CO₂ 5% should be transduced immediately. Why waiting 24h? Does the short-term effect depend on tor? It is transient? Can the effect be seen at 24h depend on something else? For instance, a compensatory mechanism. It might be important here to describe and to analyse the genes regulated after shorter time.
Response: Indeed, the delayed transcriptional response is surprising as we note. We suggest that this delay is, as the reviewer suggests, more consistent with a compensatory effect in response to physiologic and biophysical effects of CO₂ rather than to a direct sensing mechanism (line 191-193). In response to the reviewer's other queries, we have re-analyzed the 8-hour time point (the 4-hour has almost no gene changes). Interestingly, this set of genes is highly enriched in membrane and integral membrane proteins. This is consistent with data in the remainder of the paper indicating that membrane homeostasis is critical for CO₂ response. We thank the reviewer for prompting this re-examination. See lines 176-180.

5) Figure 3D: the position of the labels is misleading. The percentage analysis in the text are interesting but not the one in the figure. These last ones should be deleted.
Response: We agree and have removed them. Thank you for noting that.

6) Figure 4. Phosphorylation of MPK1: it would be easier to read if quantification of the band intensity was done. Furthermore, in the RIM101 blot, the size of the processed and unprocessed protein should be indicated by an arrow. A control using an unrelated protein should be used to make sure that this result is not due to a general protein degradation. The scheme for RIM101 might be good but should be improved (for instance, the size and the color of the peptide released from RIM101 upon proteolyze is not the same as in the preprocessed protein.
Response: A. We have included quantitation of the Mpk1 blot. B. The Rim101 processed and unprocessed proteins have been indicated. C. The total protein recovered from cells lysates generated in ambient and 5%CO₂ conditions are similar in the Mpk1 blot, supporting the conclusion that there is not general protein degradation in CO₂. D. We have altered the Rim101 scheme as suggested.

7) The selection of the 118 genes used for the nanostring experiment is not justified. Are they the most altered, the most expressed, other criteria?

Response: We agree and have added additional details behind that discussion. Line 252-254.

8) The best homolog of CNAG_07799 is not PDR5 but SNQ2 although it is not reciprocal. PDR5 is only the third best homolog. The homology is high not modest as written line 292. Although I agree that CNAG_07799 seems to ask as a Floppase and cannot be named PDR5 or SNQ2, it cannot be named neither LAT1 which is *S. cerevisiae* design a Dihydrolipoamide acetyltransferase (source SGD). This part should be changed.

Response: Please see above discussion about using PDR9 as the name.

9) Figure 6: Disk assays are good but these experiments seem to have been performed only once and not statistical analysis of the difference of the zone of clearance size is given. This should be corrected.

Response: We performed these experiments in triplicate and observed the same zone of clearance in all replicates. We have added that to the legend. We have also changed the figure to include the zone of clearance as insets rather than as bar graphs.

REVIEWERS' COMMENTS

Reviewer #1 (Remarks to the Author):

The authors have made commendable efforts in addressing the concerns raised during the previous evaluation and have presented a substantially revised manuscript accompanied by a range of new experiments. These new experiments include internalization analysis of NBD-PS, manipulation of PS levels, and localization analysis of Pdr5. While the revisions are appreciated, there are still misleading statements and overinterpretations of the new results that should be addressed in a second revision as outline below:

A significant concern that persists is the interpretation of the Annexin V binding assay as a direct indicator of phosphatidylserine (PS) exposure. It is crucial to acknowledge that there exist publications in the field demonstrating that the classical Annexin V staining technique may not be specific for PS (e.g., Yeung T, Heit B, Dubuisson JF, Fairn GD, Chiu B, et al. (2009) Contribution of phosphatidylserine to membrane surface charge and protein targeting during phagosome maturation. *J Cell Biol*: 917–928; Weingartner A, Kemmer G, Muller FD, Zampieri RA, Gonzaga dos Santos M, et al. (2012) Leishmania Promastigotes Lack Phosphatidylserine but Bind Annexin V upon Permeabilization or Miltefosine Treatment. *PLoS ONE* 7(8): e42070)). These works provide evidence that Annexin V binding can occur to anionic lipids. Regrettably, these relevant publications are not referenced within the manuscript, and the topic is not critically discussed. In light of these findings, the authors should consider rephrasing the statement regarding PS exposure. It is more accurate to state that there is evidence of anionic lipids being exposed to the plasma membrane, aligning with the author's conclusion on changes in plasma membrane lipid asymmetry.

Regarding Figure 6e, which presents data on PS flippase activity, it is essential to enhance the clarity of the presented results. The authors are encouraged to provide cellular images that visually demonstrate the intracellular accumulation of NBD-PS. This will bolster the argument that NBD-PS is genuinely internalized rather than merely associated with the plasma membrane. Moreover, to exclude degradation of NBD-PS, a thin-layer chromatography (TLC) analysis of the cell extract should be presented. This analysis would serve to validate that NBD-PS is not metabolized into a derivative at the cell surface prior to internalization. Given the authors' reporting on phosphatidylethanolamine (PE) exposure, it is recommended to complement the NBD-PS internalization data with analogous data on NBD-PE uptake.

In conclusion, the authors' efforts in revising the manuscript and incorporating new experiments are acknowledged. However, the concerns expressed in the initial evaluation remain partially unresolved. Addressing the points highlighted here will contribute to the robustness and accuracy of the conclusions drawn from the new results. This comprehensive revision will ultimately strengthen the quality and impact of the manuscript.

Reviewer #2 (Remarks to the Author):

The authors build upon a prior intriguing observation in which they noted a striking correlation between CO₂ tolerance and mammalian pathogenesis among clinical and environmental *C. neoformans* strains. In the current manuscript, they explore mechanisms by which CO₂ might be sensed, as well as microbial cell responses to 5% CO₂. They clearly identified the TOR pathway as a contributor to this response. They also build a very well-supported model in which changes in the PS concentration of the outer cell membrane play a central role in the sensing and response to environmental CO₂. This is a provocative model that will certainly generate new hypotheses and related experiments. The paper is also important in the way that it emphasizes the precision of regulation required among many signaling pathways previously implicated in the response of *C. neoformans* to the infected host. Their unexpected finding of CO₂ suppression of the activation of two other virulence-related fungal signaling pathways adds subtlety to prior work in this area.

Minor points

Line 197. The following phrase is confusing and requires revision: "Overall, the transcriptional response to CO₂ is delayed that and appears ...".

The authors have substantively addressed all issues from the prior review.

Reviewer #3 (Remarks to the Author):

The regulation of plasma membrane asymmetry by CO₂ in a TOR and PDR9 dependent pathway is strongly supported. However, the localization of PDR9 at the late Golgi suggests more of an indirect effect on potential floppase activity rather than directly functioning as a floppase protein. Specifically, the mechanism by which the floppase operates at the late Golgi requires a more comprehensive explanation than provided in lines 565-566. How would this floppase impact PM organization? How can the asymmetry of PE at the PM be maintained through its action? While it remains plausible that PDR9 could function as a floppase, its observed effects at the PM seem to lean more towards an indirect influence.

Minor:

1) I still have trouble understanding the sentence in lines 124-125. Based on the data presented in Figure 1d, the *swe102* mutant shows no fitness defect. However, it has been included in the reconstruction process due to its fitness phenotype. Despite this, the sentence suggests that the phenotype was not complemented. There appears to be some ambiguity here.

2) The figure 2 should be simplified and focused on mutants that exhibit a clear phenotype, and for which subsequent investigations have been carried out. This would help in presenting a more concise and informative visual representation.

Ristow et al – Response to Reviewers:

Introduction

We thank the reviewers for their constructive comments. We have revised the manuscript further to address these suggestions. With respect to new experiments suggested by Reviewer 1, we have included new lipid transport assay data indicating that overexpression of *PDR9* reduces uptake of NBD-PE (Fig. 6g). This result is consistent with the increased sensitivity of the *PDR9* overexpression strain to duramycin. As a further control suggested by Reviewer 1, we have also provided fluorescence microscopy data indicating that both NBD-PS and -PE (Supplementary Fig. 5A&B) are taken up into the cell (and not simply associated with the cell periphery). These data further suggest that Pdr9 functions indirectly or directly as a PE floppase. We have tried to be careful not to overstate the case for Pdr9 acting as a floppase. We continue to do so in the revised manuscript and have further revised our language as suggested by both Reviewers 1 and 3.

We have also responded to other comments and queries as detailed below.

Overall, we feel the manuscript is much stronger and thank the reviewers for their contribution to its improvement.

Reviewer #1 (Remarks to the Author):

The authors have made commendable efforts in addressing the concerns raised during the previous evaluation and have presented a substantially revised manuscript accompanied by a range of new experiments. These new experiments include internalization analysis of NBD-PS, manipulation of PS levels, and localization analysis of Pdr5. While the revisions are appreciated, there are still misleading statements and overinterpretations of the new results that should be addressed in a second revision as outline below:

A significant concern that persists is the interpretation of the Annexin V binding assay as a direct indicator of phosphatidylserine (PS) exposure. It is crucial to acknowledge that there exist publications in the field demonstrating that the classical Annexin V staining technique may not be specific for PS (e.g., Yeung T, Heit B, Dubuisson JF, Fairn GD, Chiu B, et al. (2009) Contribution of phosphatidylserine to membrane surface charge and protein targeting during phagosome maturation. *J Cell Biol*: 917–928; Weingartner A, Kemmer G, Muller FD, Zampieri RA, Gonzaga dos Santos M, et al. (2012) *Leishmania* Promastigotes Lack Phosphatidylserine but Bind Annexin V upon Permeabilization or Miltefosine Treatment. *PLoS ONE* 7(8): e42070)). These works provide evidence that Annexin V binding can occur to anionic lipids. Regrettably, these relevant publications are not referenced within the manuscript, and the topic is not critically discussed. In light of these findings, the authors should consider rephrasing the statement regarding PS exposure. It is more accurate to state that there is evidence of anionic lipids being exposed to the plasma membrane, aligning with the author's conclusion on changes in plasma membrane lipid asymmetry.

Response: Thank you for bringing those references to our attention. We have added the following section regarding the limitations of Annexin V staining to the discussion section (lines 552-562): **Although annexin V staining is widely used as an assay for alterations in PS distribution between the inner and outer leaflets of the plasma membrane, some studies have indicated that annexin V is not specific for PS and can also bind to other anionic phospholipids such**

as PE, phosphatidylglycerol and phosphatidic acid (45, 46). Our uptake assays indicate that there is decreased transport of PS into the cells in the presence of CO₂, supporting the notion that PS distribution is altered. However, it is also possible that the increased Annexin V staining observed in elevated CO₂ concentrations is due to the accumulation of other anionic phospholipids on the outer leaflet of the plasma membrane. Since duramycin sensitivity does not change at elevated CO₂, other anionic phospholipids would most likely be responsible for these effects. In either case, our data strongly support the notion that tolerance of host concentrations of CO₂ requires a significant remodeling of outer membrane anionic phospholipid distribution; for simplicity we will refer to PS in the remainder of the discussion with the understanding that other anionic phospholipids may also be involved. The references 45 and 46 are those suggested by the reviewer.

Regarding Figure 6e, which presents data on PS flippase activity, it is essential to enhance the clarity of the presented results. The authors are encouraged to provide cellular images that visually demonstrate the intracellular accumulation of NBD-PS. This will bolster the argument that NBD-PS is genuinely internalized rather than merely associated with the plasma membrane.

Response: We have added microscopy data indicating that NBD-PS uptake leads to intracellular accumulation of the labeled PS (**Supplementary Fig. 5A, line 383**).

Moreover, to exclude degradation of NBD-PS, a thin-layer chromatography (TLC) analysis of the cell extract should be presented. This analysis would serve to validate that NBD-PS is not metabolized into a derivative at the cell surface prior to internalization.

Response: NBD-PS is degraded in the vacuole by vacuolar enzymes and is further metabolized to PE by cytosolic enzymes. Therefore, NBD-PS would not be expected to be stable in cell extracts using the suggested assay. This uptake assay has been used extensively for decades in multiple species of yeast including *C. neoformans* (reference 34). Therefore, we have not performed this control and respectfully submit that there are no current methods to do so.

Given the authors' reporting on phosphatidylethanolamine (PE) exposure, it is recommended to complement the NBD-PS internalization data with analogous data on NBD-PE uptake.

Response: We have performed this experiment and the data are provided in **Fig. 6g** and the corresponding control for intracellular uptake provided in **Supplementary Fig. 5B**. These data are consistent with our duramycin disk diffusion assays indicating that overexpression of *PDR9* leads to increased outer membrane PE.

In conclusion, the authors' efforts in revising the manuscript and incorporating new experiments are acknowledged. However, the concerns expressed in the initial evaluation remain partially unresolved. Addressing the points highlighted here will contribute to the robustness and accuracy of the conclusions drawn from the new results. This comprehensive revision will ultimately strengthen the quality and impact of the manuscript.

Reviewer #2 (Remarks to the Author):

The authors build upon a prior intriguing observation in which they noted a striking correlation between CO₂ tolerance and mammalian pathogenesis among clinical and environmental *C. neoformans* strains. In the current manuscript, they explore mechanisms by which CO₂ might be sensed, as well as microbial cell responses to 5% CO₂. They clearly identified the TOR

pathway as a contributor to this response. They also build a very well-supported model in which changes in the PS concentration of the outer cell membrane play a central role in the sensing and response to environmental CO₂. This is a provocative model that will certainly generate new hypotheses and related experiments. The paper is also important in the way that it emphasizes the precision of regulation required among many signaling pathways previously implicated in the response of *C. neoformans* to the infected host. Their unexpected finding of CO₂ suppression of the activation of two other virulence-related fungal signaling pathways adds subtlety to prior work in this area.

Minor points

Line 197. The following phrase is confusing and requires revision: "Overall, the transcriptional response to CO₂ is delayed that and appears ...".

Response: Thank you for pointing out that awkward phrase. We have revised the sentence to read (line 195): **Overall, the transcriptional response to CO₂ occurs over multiple hours and appears to have two phases: an early phase that is highly enriched for membrane associated genes and a late phase that is consistent with the activation of the TOR pathway.**

The authors have substantively addressed all issues from the prior review.

Reviewer #3 (Remarks to the Author):

The regulation of plasma membrane asymmetry by CO₂ in a TOR and PDR9 dependent pathway is strongly supported. However, the localization of PDR9 at the late Golgi suggests more of an indirect effect on potential floppase activity rather than directly functioning as a floppase protein. Specifically, the mechanism by which the floppase operates at the late Golgi requires a more comprehensive explanation than provided in lines 565-566. How would this floppase impact PM organization? How can the asymmetry of PE at the PM be maintained through its action? While it remains plausible that PDR9 could function as a floppase, its observed effects at the PM seem to lean more towards an indirect influence.

Response: While we have provided additional support for a direct or indirect floppase function of Pdr9 (see above), we **agree** that it is much more likely to represent an indirect effect. As noted in the introduction, we have endeavored to be very cautious on this note given that no real bona fide floppase has ever been definitively characterized. We have added additional qualifying language to multiple sections of the manuscript to make this point as clear as we can: **1) Line 363-367: However, Pdr9 does not localize to the PM, which is where most proposed floppases are found, raising the possibility that this effect on PE distribution may be indirect. 2) Line 405-406: These data further support, but do not conclusively establish, the possible function of Pdr9 as a phospholipid floppase. 3) 586-588: Taken together, the complex effect of elevated expression of PDR9 on lipid asymmetry suggests it does so through a combination of direct and indirect effects on lipid homeostasis. 4) 595-598: Regardless of its likely indirect and complex mechanism of action toward CO₂ tolerance, the effect of PDR9 on multiple facets of lipid homeostasis and CO₂ susceptibility has provided important insights into the mechanisms of the latter.**

Minor:

1) I still have trouble understanding the sentence in lines 124-125. Based on the data presented in Figure 1d, the swe102 mutant shows no fitness defect. However, it has been included in the

reconstruction process due to its fitness phenotype. Despite this, the sentence suggests that the phenotype was not complemented. There appears to be some ambiguity here.

Response: We have deleted this sentence and the *swe102* mutant from **Fig. 1d**.

2) The figure 2 should be simplified and focused on mutants that exhibit a clear phenotype, and for which subsequent investigations have been carried out. This would help in presenting a more concise and informative visual representation.

Response: As suggested, we have reduced the strains presented in **Fig. 2** to H99, the environmental strain examined in the analysis of Mpk1 phosphorylation in Fig. 4 and the *rim101* mutant which is further studied as well.